# Myofibroblast senescence promotes arrhythmogenic remodeling in the aged infarcted rabbit heart

Brett C Baggett[1,2†], Kevin R Murphy[1,2†], Elif Sengun[1,2,3†], Eric Mi[1,2], Yueming Cao[1,2], Nilufer N Turan[2], Yichun Lu[2], Lorraine Schofield[2], Tae Yun Kim[2], Anatoli Y Kabakov[1,2], Peter Bronk[2], Zhilin Qu[4], Patrizia Camelliti[5], Patrycja Dubielecka[1,6], Dmitry Terentyev[2], Federica del Monte[7], Bum-Rak Choi[2], John Sedivy[1], Gideon Koren[1,2]*

[1]Brown University, Providence, United States; [2]Cardiovascular Research Center, Rhode Island Hospital, Providence, United States; [3]Department of Pharmacology, Institute of Graduate Studies in Health Sciences, Istanbul University, Istanbul, Turkey; [4]School of Medicine, University of California, Los Angeles, Los Angeles, United States; [5]School of Biosciences and Medicine, University of Surrey, Guildford, United Kingdom; [6]Department of Hematology, Rhode Island Hospital, Providence, United States; [7]Medical University of South Carolina, Charleston, United States

**\*For correspondence:**
gideon_koren@brown.edu

[†]These authors contributed equally to this work

**Competing interest:** The authors declare that no competing interests exist.

**Abstract** Progressive tissue remodeling after myocardial infarction (MI) promotes cardiac arrhythmias. This process is well studied in young animals, but little is known about pro-arrhythmic changes in aged animals. Senescent cells accumulate with age and accelerate age-associated diseases. Senescent cells interfere with cardiac function and outcome post-MI with age, but studies have not been performed in larger animals, and the mechanisms are unknown. Specifically, age-associated changes in timecourse of senescence and related changes in inflammation and fibrosis are not well understood. Additionally, the cellular and systemic role of senescence and its inflammatory milieu in influencing arrhythmogenesis with age is not clear, particularly in large animal models with cardiac electrophysiology more similar to humans than previously studied animal models. Here, we investigated the role of senescence in regulating inflammation, fibrosis, and arrhythmogenesis in young and aged infarcted rabbits. Aged rabbits exhibited increased peri-procedural mortality and arrhythmogenic electrophysiological remodeling at the infarct border zone (IBZ) compared to young rabbits. Studies of the aged infarct zone revealed persistent myofibroblast senescence and increased inflammatory signaling over a 12-week timecourse. Senescent IBZ myofibroblasts in aged rabbits appear to be coupled to myocytes, and our computational modeling showed that senescent myofibroblast-cardiomyocyte coupling prolongs action potential duration (APD) and facilitates conduction block permissive of arrhythmias. Aged infarcted human ventricles show levels of senescence consistent with aged rabbits, and senescent myofibroblasts also couple to IBZ myocytes. Our findings suggest that therapeutic interventions targeting senescent cells may mitigate arrhythmias post-MI with age.

## Editor's evaluation

This study describes important results and convincing evidence linking myofibroblast senescence in the aged heart with a pro-arrhythmogenic phenotype. This is in turn related to higher mortality after myocardial infarction in the aged rabbit heart. These constitute important empiric as opposed to detailed findings. They nevertheless will be of interest to clinician scientists studying cardiac function and disease.

## Introduction

Sudden cardiac death (SCD) remains the leading cause of death worldwide and is responsible for up to 20% of all deaths in the USA (*Benjamin et al., 2019*; *Deo and Albert, 2012*). The incidence of SCD increases exponentially with age, which poses a growing problem as the portion of the US population 65 years or older is estimated to increase to 15% by 2030 (*Benjamin et al., 2019*; *Partridge et al., 2018*). The vast majority of SCDs are caused by ventricular tachycardia/ventricular fibrillation (VT/VF) subsequent to myocardial infarction (MI) (*Kolettis, 2013*). Immediate clinical reperfusion interventions can limit the extent of acute ischemic injury and acute arrhythmic mortality post-MI, however progressive tissue remodeling in the days-to-years post-MI can establish a substrate and trigger for potentially lethal arrhythmias (*Mendonca Costa et al., 2018*). This tissue remodeling process is well described in young animals (*van den Borne et al., 2010*), but less is known about what changes to this process occur with age and how these changes increase risk of arrhythmia initiation and propagation.

We have previously characterized the efficacy of the New Zealand White rabbit as a model of the aging human heart. We showed that the aging rabbit heart recapitulates cardiac dysfunctions observed in aging humans, including aortic stiffening, reduced ventricular compliance, increased interstitial fibrosis, abnormal conduction in the His-Purkinje system, and lower thresholds for induced ventricular arrhythmias (*Cooper et al., 2012*). We also demonstrated defects in autophagy and mitochondrial function in isolated aged rabbit myocytes resulting in increased mitochondrial reactive oxygen species which oxidized the ryanodine receptor and promoted spontaneous $Ca^{2+}$ release (*Cooper et al., 2013*; *Murphy et al., 2019*). To better investigate the interaction between post-MI tissue remodeling and arrhythmogenesis with age, we have established a minimally invasive surgical procedure to reproducibly infarct the rabbit heart regardless of age or variable coronary anatomy, whereby an embolic coil occludes the left coronary artery (*Morrissey et al., 2017*). Infarction by this method results in electrophysiological remodeling characteristic of human MI, including functional reentry anchored at the infarct border zone (IBZ), reduction in ejection fraction, and lower thresholds for monomorphic VT induced by programmed stimulation (*Morrissey et al., 2017*; *Ziv et al., 2012*).

Cellular senescence is a stress response characterized by irreversible proliferation arrest, resistance to apoptosis, and secretion of a collection of inflammatory cytokines, growth factors, and proteases termed the senescence-associated secretory phenotype (SASP) (*Coppé et al., 2010*; *Rodier et al., 2009*; *Campisi and Robert, 2014*). Transient senescence plays beneficial roles in young animals in wound healing by limiting fibrosis, in tumor suppression, and in development (*Wang et al., 2011*; *Storer et al., 2013*; *Demaria et al., 2014*). However, senescent cells accumulate with age in many tissues and accelerate many aging-associated pathologies in a paracrine manner via SASP-mediated chronic inflammatory signaling and/or in a juxtacrine manner via transfer of intracellular proteins to neighboring cells (*Prata et al., 2018*; *Vicente et al., 2016*; *Burton and Krizhanovsky, 2014*; *Shibamoto et al., 2019*; *Kirschner et al., 2020*; *Biran et al., 2015*). Genetic or pharmacological elimination of senescent cells with age slows the onset and progression of many aging pathologies, including atherosclerosis, diabetes, idiopathic pulmonary fibrosis, osteoarthritis, and neurodegenerative diseases (*Baker et al., 2016*; *Baker et al., 2011*; *Childs et al., 2015*; *Zhu et al., 2015*; *Chang et al., 2016*; *Kim and Kim, 2019*; *Kirkland et al., 2017*; *van Deursen, 2019*; *Kirkland and Tchkonia, 2017*).

In the adult infarcted wild-type mouse heart, senescent cells arise in the scar and IBZ around day 3 post-MI and are largely cleared by day 7 (*Shibamoto et al., 2019*; *Zhu et al., 2013*). Most of these senescent cells are cardiac myofibroblasts, the main cell type responsible for secreting extracellular matrix components forming the scar. This observation is consistent with epidermal wound studies in mice, suggesting a role for transient myofibroblast senescence as a normal part of the wound healing process in limiting fibrosis (*Demaria et al., 2014*). Infarction of adult constitutive p53 KO mice, in which senescence induction is impeded, results in increased fibrosis and decreased inflammation in the scar 7 days post-MI (*Zhu et al., 2013*). In a cell culture model of isolated mouse cardiac fibroblasts, senescent cardiac fibroblasts limit the proliferation of non-senescent cardiac fibroblasts in a juxtacrine but not paracrine fashion (*Shibamoto et al., 2019*). These results suggest a beneficial role for transient myofibroblast senescence in limiting fibrosis post-MI. Conversely, insights toward a pathological role of chronic cardiac senescence come from aging mouse models of MI. Treatment of aged mice with the senolytic drug navitoclax (i.e. a drug that specifically eliminates senescent cells) before or after MI significantly reduces senescence burden, improves cardiac function, mitigates cardiac remodeling, and reduces scar size post-MI, although the effects on arrhythmias has not been studied to our

knowledge (*Dookun et al., 2020*). These results suggest that acute induction of senescence followed by timely clearance limits excess scarring, but persistence of cardiac senescence as seen with age can promote cardiac fibrogenesis and worsen cardiac function and might lead to increased risk of arrhythmias. Although these findings in mice are important, the mechanisms underlying a relationship between age-associated senescence and arrhythmogenesis is not well understood, particularly in an animal model like the rabbit whose cardiac electrophysiology is more functionally relevant to humans than that of mice.

Potential pro-arrhythmic effects of senescent cells could occur through a paracrine fashion via SASP components. Indeed, treatment of isolated rat and mouse cardiomyocytes with exogenous SASP components including IL-6, IL-1β, and TNF-α have pro-arrhythmic effects in ion channel remodeling (*Francis Stuart et al., 2016*; *Aromolaran et al., 2018*). Senescent myofibroblasts might be able to interfere with cardiomyocyte electrophysiology from relatively long distances through chronic inflammatory signaling. Alternatively, the direct cell-cell coupling via gap junctions of senescent myofibroblasts to cardiomyocytes at the IBZ might alter their electrophysiology more than coupling with non-senescent myofibroblasts. Such a process would establish regional heterogeneities in ion channel activity, action potential duration (APD), and other electrophysiological factors anchored at the IBZ that would permit reentrant current and therefore VT/VF.

We hypothesized that aged infarcted rabbits experience a persistence of senescent myofibroblasts in the scar and IBZ compared to young, and that these senescent myofibroblasts act in a paracrine or juxtacrine fashion to induce pro-arrhythmic remodeling in IBZ cardiomyocytes. Here, we demonstrate that aged rabbits compared to young exhibit increased peri-procedural deaths mostly due to VT/VF, consistent with prolongation of APD and alternans at the IBZ associated with higher frequency VF. We observed no difference in the size of the scar or IBZ geometry over the first 12 weeks post-MI between young and aged rabbits. However, whereas senescent cells in the young rabbit scar were largely cleared by 3 weeks post-MI, senescence of mostly myofibroblasts remained high in the aged rabbits up to 12 weeks post-MI and correlated with increased local expression of inflammatory cytokines. In ventricular tissue samples from aged infarcted human patients, we observed elevated levels of senescence markers, and most senescent cells appeared to be myofibroblasts. Using primary adult rabbit

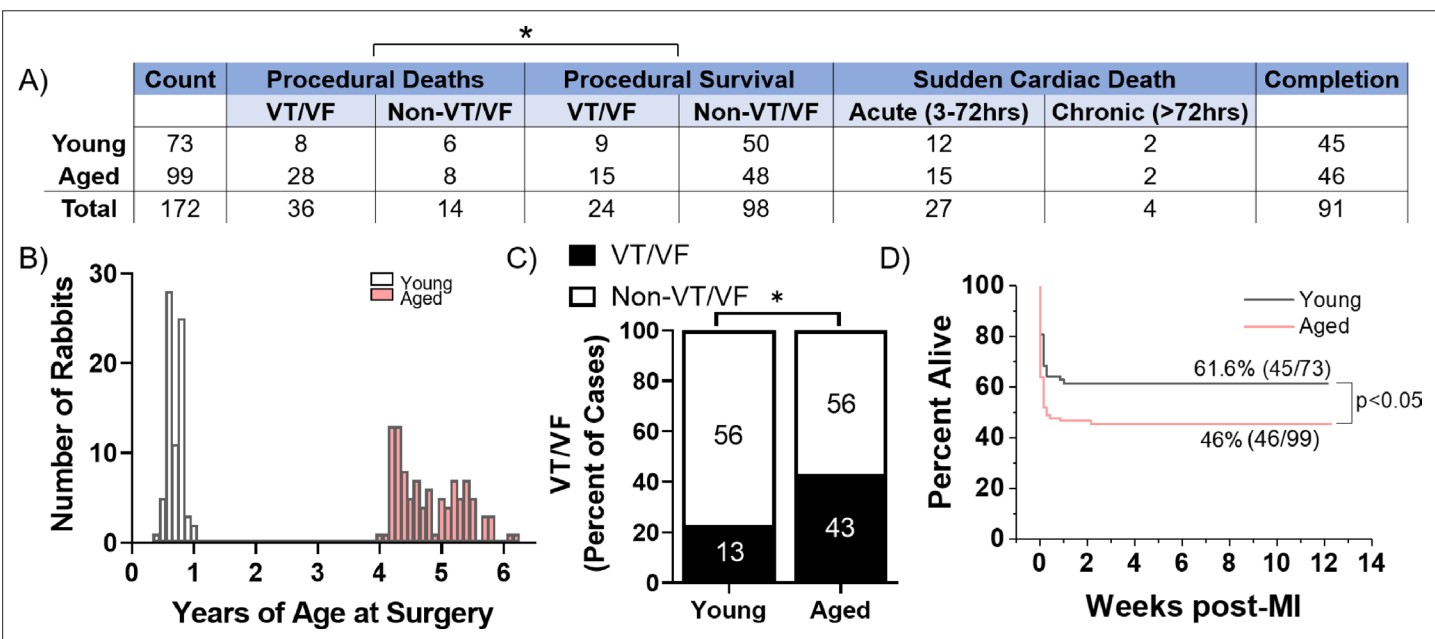

**Figure 1.** Aged rabbits exhibit increased incidence of peri-procedural arrhythmias. (**A**) Survival table of young and aged infarcted rabbits. Procedural deaths were defined as death occurring within the first 3 hr of surgery. (**B**) Histogram of rabbits included in the study by age at the time of surgery. (**C**) Incidence of procedural ventricular tachycardia/ventricular fibrillation (VT/VF) in young and aged infarcted rabbits. Numbers inside bars are number of rabbits. *p<0.05, two-tailed exact test. (**D**) Survival curves of young and aged rabbits post-MI (p<0.05, log rank test).

The online version of this article includes the following source data for figure 1:

**Source data 1.** Raw data pertaining to rabbits used in the study that was used to create *Figure 1*.

cardiac fibroblasts, we induced senescence via treatment with the drug etoposide. With this method, we did not observe any effect of exogenous or co-cultured conditioned media from senescent cardiac fibroblasts on the $I_{Kr}$ current or APD of treated rabbit myocytes compared to conditioned media from proliferating cardiac fibroblasts. However, in both aged rabbit and human cardiac tissue, we observed senescent cardiac myofibroblasts in close proximity to IBZ myocytes, with the gap junction protein Cx43 appearing to couple the cells. Previous studies in rabbits have demonstrated a role for Cx43 in the coupling of myocytes and fibroblasts as evidenced by dye transfer and optogenetic studies (*Camelliti et al., 2004a*; *Camelliti et al., 2004b*; *Schultz et al., 2019*). Based on our electrophysiological and volumetric measurements of senescent and non-senescent fibroblasts, we performed computational modeling which suggested that coupling of myocytes with senescent cardiac myofibroblasts would result in prolongation of APD and lower thresholds for conduction block. Altogether, our results indicate a pathological role for the persistence of senescent cardiac myofibroblasts in directly exacerbating arrhythmogenesis at least via a direct cell-cell interaction through gap junctions with age post-MI. These findings suggest that targeted elimination of senescent cells post-MI could be an effective therapeutic method to mitigate senescence burden and combat arrhythmias in aged infarcted individuals.

## Results

### Aged infarcted rabbits exhibit increased incidence of peri-procedural arrhythmias

A total of 73 young (≤1 year) and 99 aged (≥4 years) female New Zealand White rabbits were subjected to minimally invasive coil embolization of the left coronary artery to induce MI of the apical left ventricular free wall (*Figure 1*). The average age for young rabbits was 7.8 months and for aged rabbits was 4.8 years. Aged rabbits exhibited significantly higher peri-procedural mortality compared to young rabbits (36% of aged rabbits versus 19% of young rabbits, p=0.0120), with most young and aged peri-procedural deaths attributed to VT/VF. We observed no significant difference in SCD in the acute (3–72 hr) or chronic (≥72 hr) periods post-MI between young and aged rabbits. Aged rabbits exhibited a significantly higher incidence of peri-procedural lethal or nonlethal VT/VF (44% of aged rabbits versus 23% of young rabbits, p=0.0040) (*Figure 1C*). Kaplan-Meyer analysis shows aged rabbits had a significantly decreased overall survival post-MI compared to young (p=0.0285), and all mortality of young and aged rabbits occurred within the first 48 hr post-MI (*Figure 1D*). Of rabbits that initiated peri-procedural VT/VF, we did not find any significant difference in the number or success rate of defibrillation attempts to restore sinus rhythm (p=0.38 and p=0.54, respectively, data not shown). Overall, our findings indicate the aged rabbit heart is more susceptible to acute, potentially lethal ischemic arrhythmias compared to young rabbits. These trends correlate with the age-associated increase in mortality of out-of-hospital first-time acute MI in humans (*Benjamin et al., 2019*).

### Progression of infarct size, IBZ geometry, and fibrosis is consistent between young and aged rabbits

Long-term progressive electrophysiological tissue remodeling at the IBZ can facilitate the initiation and propagation of arrhythmias post-MI even after the ischemic event, increasing risk of SCD (*Axford-Gatley and Wilson, 1988*). Larger scars and increased IBZ fibrosis can establish a source-sink mismatch and regional heterogeneities in APD and conduction velocity which underlie VT/VF post-MI (*Richardson et al., 2015*; *Neuschl et al., 2018*). Additionally, progressive infarct expansion can result in declining cardiac function and increased risk of ventricular wall rupture or VT/VF (*Richardson and Holmes, 2015*; *Zhang et al., 2014*). However, the cellular mechanisms underlying age-associated differences in post-MI tissue remodeling and their arrhythmogenic consequences are not well understood. We previously demonstrated that coil embolization of the young and aged rabbit left coronary artery resulted in similar scar size 3 weeks post-MI (*Morrissey et al., 2017*). Here, we further investigated potential age-associated changes in the dynamics of scar size, shape, and fibrosis throughout a 12-week timecourse post-MI. Cardiac tissue was harvested from young and aged infarcted rabbits at 1, 2, and 3 weeks post-MI to encompass the period following scar formation, and at 12 weeks post-MI to interrogate potential long-term changes.

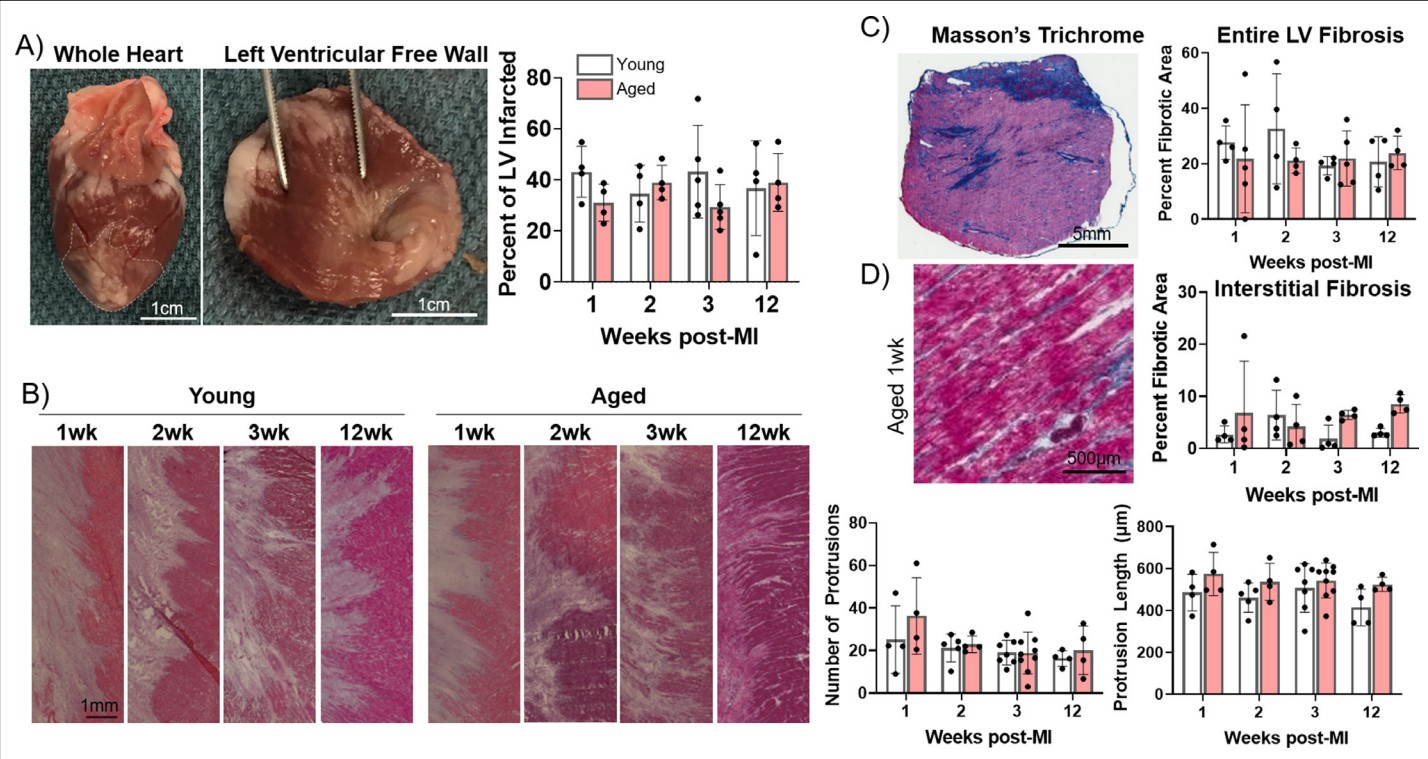

**Figure 2.** Progression of infarct size, infarct border zone (IBZ) geometry, and fibrosis is consistent between young and aged rabbits. (**A**) Left: Whole rabbit heart, with infarct zone outlined. Middle: Dissected left ventricular free wall. Right: Quantification of percent infarcted area of the left ventricular free wall. (**B**) Left: Representative hematoxylin and eosin (H&E)-stained images of IBZ (left portion of images) protrusions into surviving myocardia (right portion of images). Right: Quantification of number (left) and length (right) of protrusions. (**C**) Left: Representative Masson's trichrome-stained left ventricular section. Right: Quantification of percent fibrotic area. (**D**) Left: Representative Masson's trichrome-stained remote zone images showing interstitial fibrosis. Right: Quantification of % fibrotic interstitial area. Dots represent average data for each rabbit, error bars SEM.

The online version of this article includes the following source data for figure 2:

**Source data 1.** Raw data used to create *Figure 2*.

To assess scar size, we measured the percent infarcted area from epicardial left ventricular free wall dissections via gross anatomic photos. We found no significant difference in the percent infarcted area of the left ventricle between young and aged rabbits at any timepoint (*Figure 2A*). We then compared the functional area of the IBZ between young and aged infarcted rabbits as represented by protrusions of the scar into the surviving myocardia, since IBZ geometry can affect regional heterogeneities in conduction velocity and APD. We utilized hematoxylin and eosin (H&E)-stained frozen left ventricle sections from young and aged infarcted rabbit hearts, laid flat along the circumferential-longitudinal plane prior to freezing. To better characterize the 3D structure of the IBZ from each heart, at least five slides over at least a 500 µm endocardial-to-epicardial span per rabbit were analyzed. We found no significant difference in either the number or length of protrusions between young and aged rabbits at any timepoint post-MI or over time for each age group (*Figure 2B*). To assess potential age-associated changes in fibrosis post-MI, we analyzed frozen left ventricular sections stained with Masson's trichrome. We observed no difference at any timepoint between young and aged rabbits post-MI in either percent fibrotic area of the entire left ventricular free wall or in percent interstitial fibrotic area in the remote zone (RZ) (*Figure 2C–D*). Overall, we observed no significant age-associated differences in the dynamics of scar size, IBZ geometry, or fibrosis in the first 12 weeks post-MI, suggesting the physical properties of the scar are not sufficient to explain age-associated differences in arrhythmogenesis post-MI.

## Electrophysiological remodeling in the IBZ of aged rabbit hearts

Although aging is associated with changes in numerous biological processes potentially impacting infarct healing, less is known about the role of age-associated changes in post-MI electrophysiological

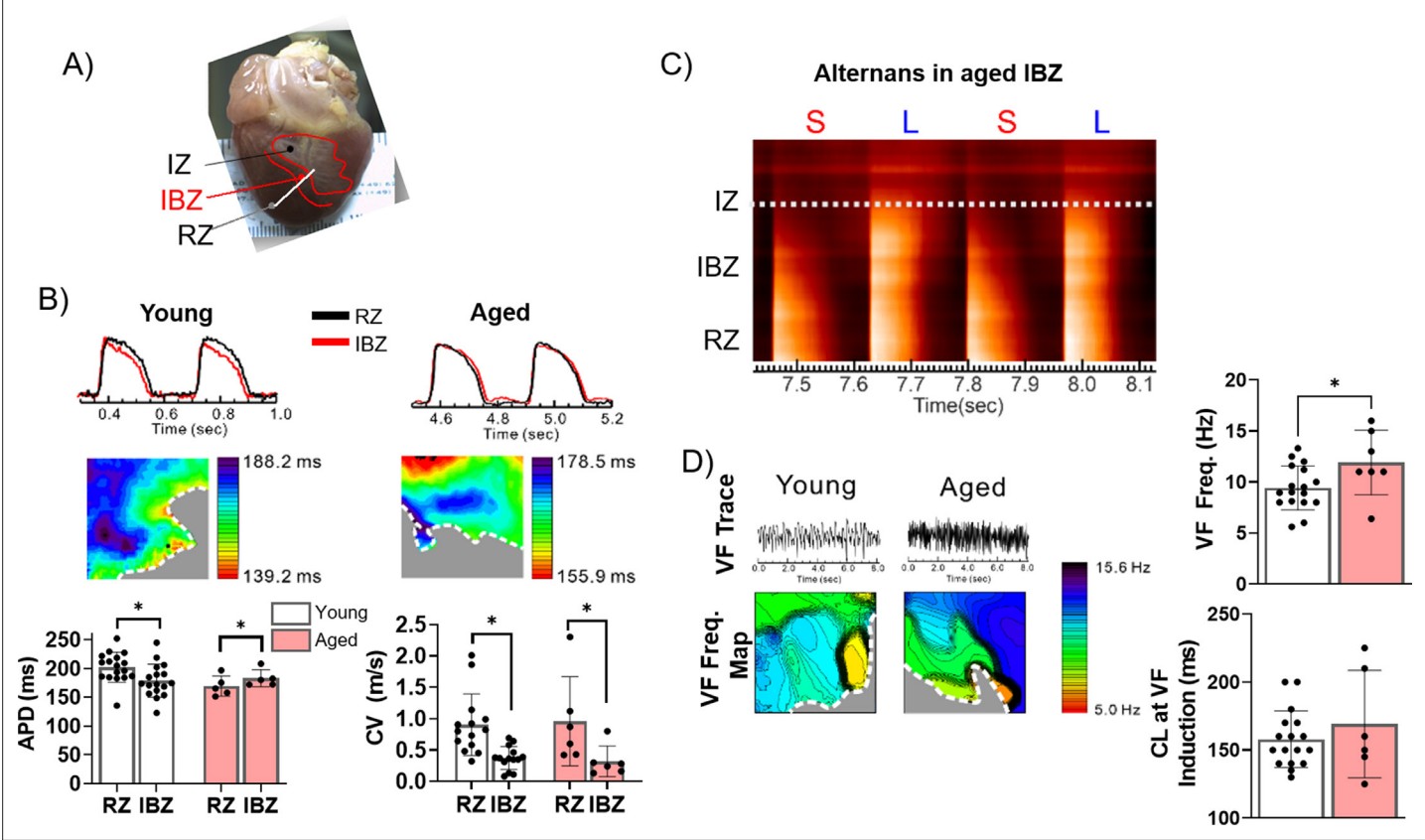

**Figure 3.** The border zone of aged rabbits shows action potential duration (APD) prolongation, APD alternans, and faster ventricular fibrillation (VF) frequency. (**A**) Photograph of a rabbit heart showing the infarct zone (IZ), infarct border zone (IBZ), and remote zone (RZ). White line indicates a representative plane along which alternans images in (**C**) were recorded. (**B**) Top: Representative action potential (AP) traces from the RZ and IBZ of young and aged rabbits. Middle: Representative APD maps from young and aged rabbits, showing APD at different points along the border zone. Bottom: Quantification of APD and conduction velocity in the RZ and IBZ of young and aged rabbits. *p<0.05, two-tailed exact test. (**C**) Representative AP trace of an aged infarcted rabbit at 3 weeks post-MI showing alternans in the IBZ. (**D**) Top: Representative traces showing VF in young and aged rabbits after electrical induction. Bottom: Representative VF frequency maps of IBZ of young and aged rabbits. Right: Quantification of VF frequency and cycle length at which VF was induced from young and aged rabbits. *p<0.05, two-tailed exact test. Dots represent average data for each rabbit. Error bars: SEM.

The online version of this article includes the following source data for figure 3:

**Source data 1.** Raw data used to create *Figure 3*.

tissue remodeling and how such changes might contribute to differential risk of lethal arrhythmias. We hypothesized that more severe electrophysiological remodeling in the aged IBZ compared to young establishes a greater arrhythmic substrate in aged rabbit hearts. To test this hypothesis, we investigated action potential (AP) dynamics in young (n=17) and aged (n=5) rabbit hearts ex vivo at 3 weeks post-MI using optical mapping. The epicardial APDs were recorded using the fluorescent voltage-sensitive dye di-4-ANEPPS and averaged from multiple randomized 4 cm² regions of the scar, IBZ, and RZ during 350 ms cycle length stimulation (*Figure 3A*). Representative AP traces and APD maps from young and aged rabbit hearts are shown in *Figure 3B*.

In aged rabbits, APD at the IBZ was significantly longer than in the RZ (183.1±6.64 ms in IBZ and 169.4±7.90 ms in RZ, p<0.05), whereas in young rabbits, APD in the IBZ was significantly shorter compared to the RZ (178.9±6.99 ms in IBZ and 202.2±6.32 ms in RZ, p<0.05). Compared to young rabbits, aged rabbits exhibited greater APD heterogeneity along the IBZ (*Figure 3B*). In both young and aged rabbits, conduction velocity was significantly slower in the IBZ compared to the RZ (0.32±0.10 m/s in IBZ and 0.96±0.30 m/s in RZ of aged hearts, 0.37±0.05 m/s in IBZ and 0.90±0.13 m/s in RZ of young hearts), consistent with previous findings (*Mendonca Costa et al., 2018*). We observed that the IBZ of aged rabbit hearts can support spatially discordant APD alternans at relatively slow heart rates, which

has been associated with an increased risk for arrhythmias (*Figure 3C*; *Liu et al., 2018*). These results were not observed in the young hearts under the same conditions. To further study age-associated changes in arrhythmogenic substrate, we induced VF by burst electrical stimulations. Representative VF traces and VF frequency maps are shown in *Figure 3D*. Aged rabbits exhibited significantly higher dominant VF frequency compared to young, which may be related to the underlying complexity of arrhythmia. Additionally, the aged rabbit hearts displayed greater spatial variation of VF frequencies across the RZ compared to young, whereas in young rabbits slower-frequency regions tended to cluster near the IBZ. Altogether, these findings suggest that age-associated changes in electrical remodeling 3 weeks post-MI increased the heterogeneity of rate-dependent APD dynamic properties between the IBZ and RZ, notably the prolongation in APD in the IBZ of aged rabbits, which demonstrates a more arrhythmogenic substrate.

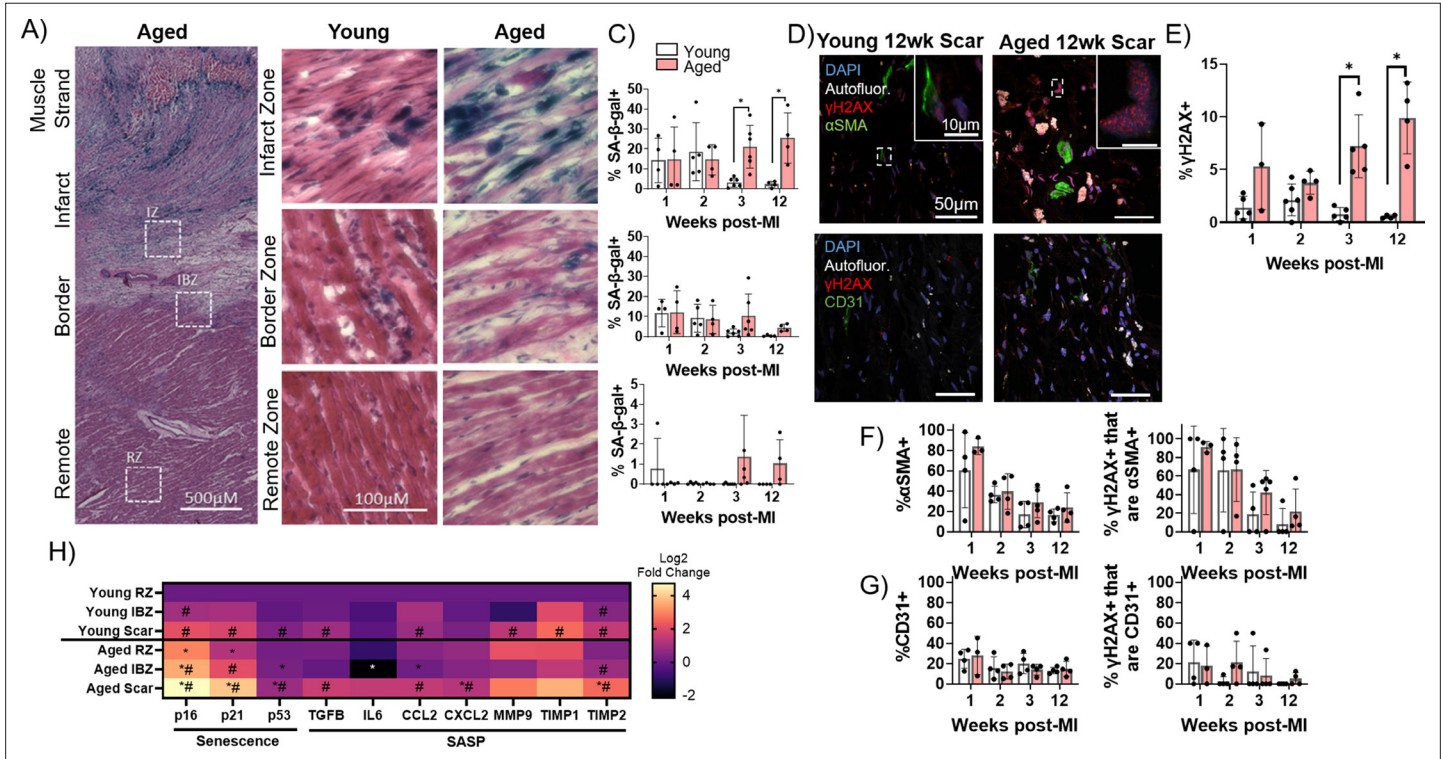

**Figure 4.** Senescence of myofibroblasts is elevated and persistent in the aged rabbit heart post-MI and correlates with increased inflammation. (**A**) Representative senescence-associated β-galactosidase (SA-β-gal)-stained image showing examples of infarct zone, border zone, and remote zone (RZ). (**B**) Representative SA-β-gal-stained images showing infarct zone, border zone, and RZ from young and aged rabbits at 3 weeks post-MI. (**C**) Quantification of percent SA-β-gal+ cells from the scar (top), infarct border zone (IBZ) (middle), and RZ (bottom) in young and aged rabbits at 1, 2, 3, and 12 weeks post-MI. (**D**) Representative confocal images of αSMA/γH2AX double immunofluorescence staining (top row) and CD31/γH2AX double immunofluorescence staining (bottom row) from young (left) and aged (right) in the infarct zone of rabbits at 12 weeks post-MI. White indicates autofluorescence and was used to avoid false positive fluorescence signal. (**E**) Quantification of % of nuclei with three or more γH2AX foci. (**F**) Quantification of percent αSMA+ cells (left) and the percent of γH2AX+ cells that are αSMA+ (right). (**G**) Quantification of percent CD31+ cells (left) and the percent of γH2AX+ cells that are CD31+ (right). (**H**) Quantification of expression of senescence and senescence-associated secretory phenotype (SASP) genes via RT-qPCR from young and aged rabbits 3 weeks post-MI. N=3 rabbits per condition. Dots represent average data for each rabbit, error bars SEM. Two-tailed exact test: *p<0.05 compared to young, # p<0.05 compared to respective RZ.

The online version of this article includes the following source data and figure supplement(s) for figure 4:

**Source data 1.** Raw data used to create *Figure 4*.

**Figure supplement 1.** Senescence assessment of young and aged sham-infarcted rabbits.

**Figure supplement 1—source data 1.** Raw data used to create *Figure 4—figure supplement 1*.

## Senescence of myofibroblasts is elevated and persistent in the aged rabbit heart post-MI and correlates with increased inflammation

To investigate potential cellular mechanisms underlying the increase in arrhythmogenic substrate in aged infarcted rabbits, we next investigated the dynamics of cellular senescence in young and aged rabbit hearts over time post-MI. Whereas in the proliferative phase post-MI, the scar is rich with myofibroblasts secreting ECM components, in the maturation phase, the fibrogenic activity of myofibroblasts is resolved at least partially via induction of senescence (*Zhu et al., 2013*; *Fu et al., 2018*; *Kanisicak et al., 2016*; *Prabhu and Frangogiannis, 2016*; *Jun and Lau, 2018*). To interrogate the dynamics of senescence in young and aged infarcted rabbit hearts, we first performed senescence-associated β-galactosidase (SA-β-gal) staining of frozen left ventricular sections from young and aged rabbits over the first 12 weeks post-MI (*Figure 4A–C*). In the aged rabbit scar, the percent of SA-β-gal+ cells in the scar remained elevated over time even at 12 weeks post-MI. Conversely in the young rabbit scar, the percent of SA-β-gal+ cells was relatively high for the first 2 weeks and largely resolved by the third week post-MI, resulting in significantly decreased senescence at 3 and 12 weeks post-MI compared to aged rabbits (*Figure 4C*). In the IBZ, the same trend was observed in young and aged rabbits but did not reach statistical significance. The young and aged rabbit RZ displayed relatively very few SA-β-gal+ cells. These RZ data are consistent with our previous assessments of baseline levels of senescence in the non-infarcted young and aged rabbit myocardia, in which we observed similarly low levels of SA-β-gal (not shown). To determine if the observed age-associated difference in senescence could be explained as an artifact of the MI procedure, we compared SA-β-gal histology between young and aged rabbits 2 weeks after sham infarction, a timepoint at which SA-β-gal signal was elevated in the young and aged infarcted rabbit scar. In the young and aged sham-infarcted rabbits, we observed low levels of SA-β-gal+ cells, with no significant difference between them (*Figure 4—figure supplement 1*).

To further characterize senescence and determine the cellular identity of senescent cells in the young and aged rabbit scar, we performed immunofluorescence staining on frozen left ventricular sections for the DNA damage marker γH2AX as a marker of senescence (*Ito et al., 2018*) and either the myofibroblast marker αSMA (*Tallquist, 2020*) or the endothelial cell marker CD31 (*Pusztaszeri et al., 2006*; *Figure 4D*). To account for transient DNA damage in otherwise healthy cells represented by low numbers of γH2AX nuclear foci, only nuclei exhibiting three or more γH2AX foci were scored as γH2AX positive ('γH2AX+)'. These results corroborated our SA-β-gal data as the percent of γH2AX+ nuclei in the aged rabbits remained elevated over time, whereas the percent of γH2AX+ nuclei in the young rabbit scar was resolved by 3 weeks post-MI, resulting in significant increases in the percent of γH2AX+ nuclei at 3 and 12 weeks post-MI in aged rabbits compared to young (*Figure 4E*). Specifically, in aged rabbits 12 weeks post-MI, we observed 9.9 ± 3% of cells were γH2AX whereas in young rabbits 12 weeks post-MI, only 0.6 ± 0.2% of cells were γH2AX+. From our αSMA immunostaining, in both the aged and young rabbit scar, we observed initially high percentages of αSMA+ cells followed by a decline over time post-MI with no significant difference between young and aged at any timepoint (*Figure 4F*, left). At 2 weeks post-MI, 39.5 ± 17.5% of cells were αSMA+ in aged rabbits and 36.5 ± 8.1% of cells were αSMA+ in young rabbits. This observation is consistent with previous mouse genetic lineage tracing reports showing fibroblasts that became myofibroblasts remained in the scar but lost αSMA expression by 2 weeks post-MI (*Fu et al., 2018*). From our double immunofluorescence staining, we observed that the majority of γH2AX+ cells were αSMA+ at 1 week post-MI in aged and young rabbits, consistent with the high levels of αSMA signal (*Figure 4F*, right). Interestingly, although at 2 weeks post-MI we observed an ~50% decrease in the percent of αSMA+ cells, we observed that ~80% of γH2AX+ cells were αSMA+, indicating a propensity toward senescence of myofibroblasts. Specifically, at 2 weeks post-MI, 66.9 ± 34.3% of γH2AX+ cells were αSMA+ in aged rabbits, and 66.2 ± 45.0% of γH2AX+ cells were αSMA+ in young rabbits. At 3 and 12 weeks post-MI, relatively few γH2AX+ cells were αSMA+, consistent with the decrease in αSMA signal. We observed no significant differences in the percent of γH2AX+ cells that were αSMA+ between young and aged at any timepoint. From our analysis of CD31 immunofluorescence staining, we observed that relatively few cells (~10–20%) were CD31+ at all timepoints with no significant difference between young and aged rabbits at any timepoint (*Figure 4G*, left), consistent with previous studies of cell composition post-MI (*Pinto et al., 2016*). At the 2 week post-MI timepoint, 12.1 ± 7.2% of cells were CD31+ in aged rabbits and 15.8 ± 11.0% of cells were CD31+ in young rabbits. Similarly, we observed that relatively

few γH2AX+ cells were CD31+ across all timepoints with no significant difference between young and aged rabbits at any timepoint, suggesting endothelial cells make up a minor percentage of senescent cells present in the scar post-MI (*Figure 4G*, right). Two-weeks post-MI, we observed 21.0 ± 20.1% of γH2AX+ cells were CD31+ in aged rabbits and 2.3 ± 4.5% of γH2AX+ cells were CD31+. These findings suggest that myofibroblasts make up the majority of persistent senescent cells in the scar of young and aged rabbits.

We next assessed the expression of senescence and SASP transcripts from flash-frozen dissections of the scar, IBZ, and RZ of young and aged rabbits at 3 weeks post-MI, a timepoint at which significant differences in senescence were observed between young and aged rabbits. In the aged scar compared to the young scar, we observed a significant upregulation in the expression of all three assayed senescence genes (p16, p21, and p53), as well as the SASP components CXCL2, and TIMP2 (*Figure 4H*). For both young and aged rabbits, in the scar compared to their respective RZ, the expression of nearly all assayed senescence and SASP genes was significantly upregulated, except IL-6, MMP9, and TIMP1 for aged rabbits and CXCL2 for young rabbits. In the aged IBZ compared to the young IBZ, we found a significant upregulation in the expression of p16 and p53, as well as a significant decrease in the expression of IL-6 and CCL2. Conversely, in the young and aged IBZ relative to their respective RZ, only p16, p21, and TIMP2 were expressed significantly higher in aged and only p16 and TIMP2 were expressed significantly higher in young. In the aged RZ compared to the young RZ, we found a significant increase in the expression of p16 and p21, with no significant decrease in expression of any genes. Overall, our data indicate at 3 weeks post-MI, expression of many senescence and SASP factors is upregulated in the aged scar and IBZ compared to young rabbits, consistent with our histological senescence data.

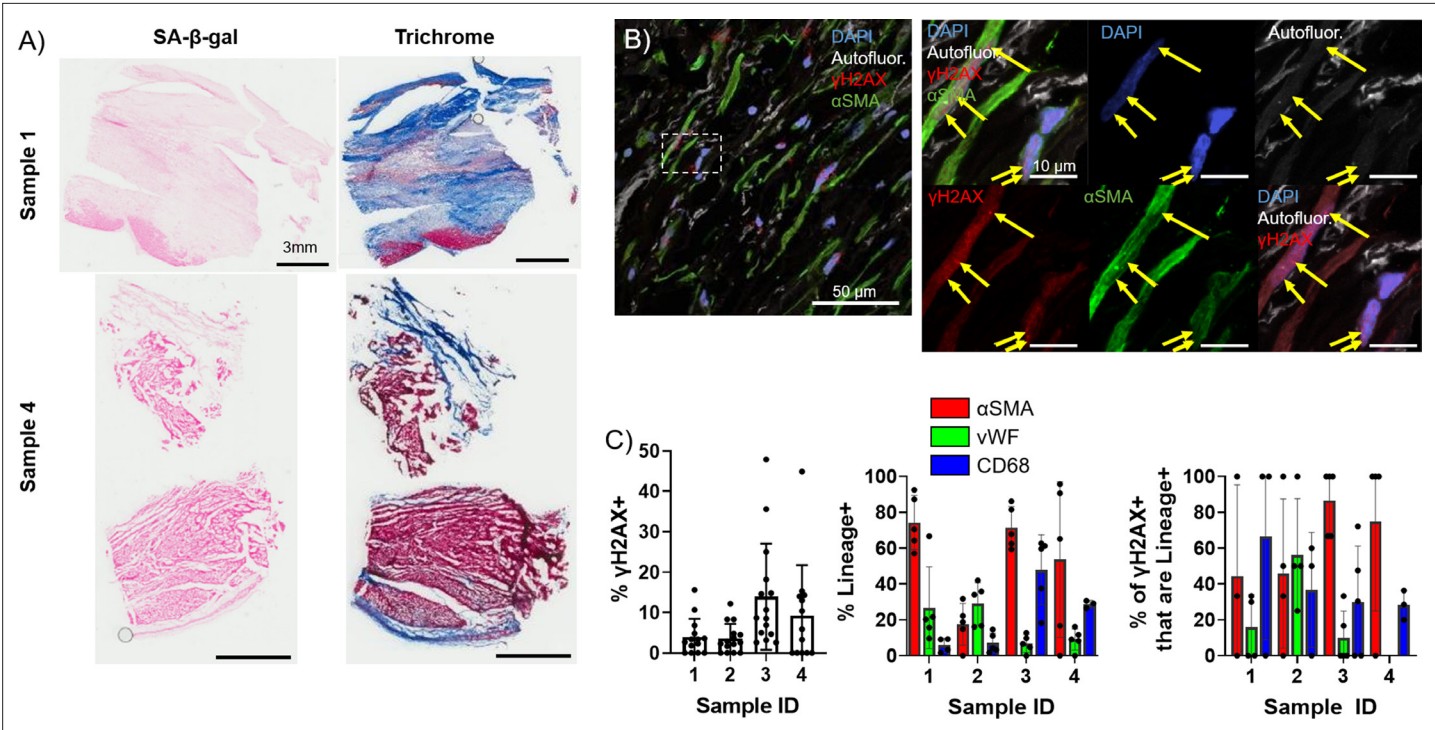

**Figure 5.** Presence of senescent myofibroblasts in aged infarcted human ventricular tissue. (**A**) Representative images of senescence-associated β-galactosidase (SA-β-gal) (left) and trichrome (right)-stained frozen sections. (**B**) Representative confocal images of γH2AX/αSMA double immunostained frozen sections. Yellow arrows indicate nuclear γH2AX foci. (**C**) Quantification of % γH2AX+ cells (left), cell identity markers (middle), and % of γH2AX+ positive for cell identity markers (right) from (B). Dots represent average data for multiple scans imaged, error bars SEM.

The online version of this article includes the following source data and figure supplement(s) for figure 5:

**Figure supplement 1.** Presence of senescent macrophages in aged infarcted human ventricular tissue.

**Source data 1.** Raw data used to create *Figure 5*, panel C.

## Presence of senescent myofibroblasts in aged infarcted human ventricular tissue

To investigate whether the observed elevation and persistence of senescence in the aged rabbit scar is recapitulated in humans, we quantified and characterized senescence in left ventricular cadaveric tissue samples from four aged male human patients with prior MI. The average patient age was 62±5 years, three of four patients presented with comorbidities, and heart failure was observed in all patients as represented by the average 2D ejection fraction of 16 ± 5%. Tissue samples flash frozen after harvest were embedded in optimal cutting temperature (OCT) prior to cryosectioning. Representative SA-β-gal and Masson's trichrome-stained frozen sections are shown in *Figure 5A*. Because very little SA-β-gal signal was observed, likely due to loss of β-galactosidase enzymatic activity in the interval between the initial tissue harvest, the flash freezing of tissue, and embedding in OCT, we investigated γH2AX foci by immunofluorescence to quantify senescence. We performed immunofluorescence staining for γH2AX, αSMA, the endothelial cell marker vWF (*Pusztaszeri et al., 2006*), and the macrophage marker CD68 (*Holness and Simmons, 1993*) in the scar. On average, we observed 8 ± 5% of total cells were γH2AX+ (*Figure 5B–C*), and from our cell identity immunofluorescence analysis we observed an average of 61 ± 30% of cells were αSMA+, 17 ± 12% of cells were vWF+, and 21 ± 20% of cells were CD68+. From our double immunofluorescence staining, we observed an average of 69 ± 28% of γH2AX+ cells were αSMA+, 20 ± 25% of γH2AX+ cells were vWF+, and 41 ± 17% of γH2AX+ cells were CD68+ (*Figure 5C*, *Figure 5—figure supplement 1*). Overall, our analysis of the human scar reveals comparable levels of cellular senescence with that seen in the aged rabbits at 3 and 12 weeks post-MI, with around 60% of senescent cells appearing to be myofibroblasts, similar to observations in the rabbit heart. These findings suggest that senescence-driven pro-arrhythmic remodeling occurring over time in aged infarcted rabbits might also occur in humans.

## Establishment of an in vitro senescence model of adult rabbit cardiac fibroblasts

To investigate how senescence might promote arrhythmogenesis with age, we first established a method to induce senescence in rabbit primary cardiac fibroblasts in vitro. Primary adult rabbit cardiac fibroblasts were isolated from adult female non-infarcted rabbits, and after two passages all cells had homogeneous myofibroblast-like characteristics indicated by immunofluorescence staining positive for αSMA and negative for the endothelial markers CD31 and vWF (*Figure 6—figure supplement 1*).

Pharmacological induction of senescence is commonly achieved in other cell culture models by treatment with DNA damaging drugs such as etoposide which induce double strand breaks (*Ito et al., 2018*; *Dai et al., 2017*). To determine whether etoposide treatment induces senescence in rabbit cardiac fibroblasts, cells were treated with either vehicle or 10 μM, 20 μM, or 40 μM etoposide and were then stained for SA-β-gal at various timepoints post-treatment (*Figure 6A–B*). Untreated, proliferating cells were also stained to assess baseline senescence. An average of 3.9 ± 1.4% of untreated, proliferating cells were SA-β-gal+, and we found no significant difference in the percent of SA-β-gal+ cells in the vehicle group at any timepoint (*Figure 6C*). For all doses of etoposide, we observed an increase in the frequency of SA-β-gal+ cells over time. Maximal SA-β-gal signal at day 12 was 73.9 ± 13.4% for the 10 μM etoposide group, 86.8 ± 4.9% for the 20 μM etoposide group, and 85.8 ± 1.8% for the 40 μM etoposide group. As well described in other in vitro models, all proliferation of cells treated with any dose of etoposide ceased upon treatment (data not shown) (*Campisi and Robert, 2014*).

In addition to SA-β-gal, we performed immunofluorescence staining against γH2AX on rabbit cardiac fibroblasts treated with vehicle or 10 μM, 20 μM, or 40 μM etoposide for 6 or 12 days, along with proliferating, untreated cells (*Figure 6D*) to assess senescence induction. Proliferating cells exhibited on average 1.23±1.0 nuclear γH2AX foci, and we found no significant change in the number of γH2AX foci per nucleus for vehicle-treated cells at any timepoint (*Figure 6E*). Cells treated with any dose of etoposide exhibited an increase in the number of γH2AX foci per nucleus over time, reaching a maximum at 12 days of treatment of 19.0±9.9 γH2AX nuclear foci for the 10 μM etoposide group, 10.5±7.8 γH2AX foci for the 20 μM etoposide group, and 16.8±15.1 γH2AX foci for the 40 μM etoposide group. In untreated, proliferating cells, an average of 6.5 ± 7.7% of cells exhibited more than five γH2AX nuclear foci, whereas after 12 days of treatment with 10 μM, 20 μM, and 40 μM etoposide,

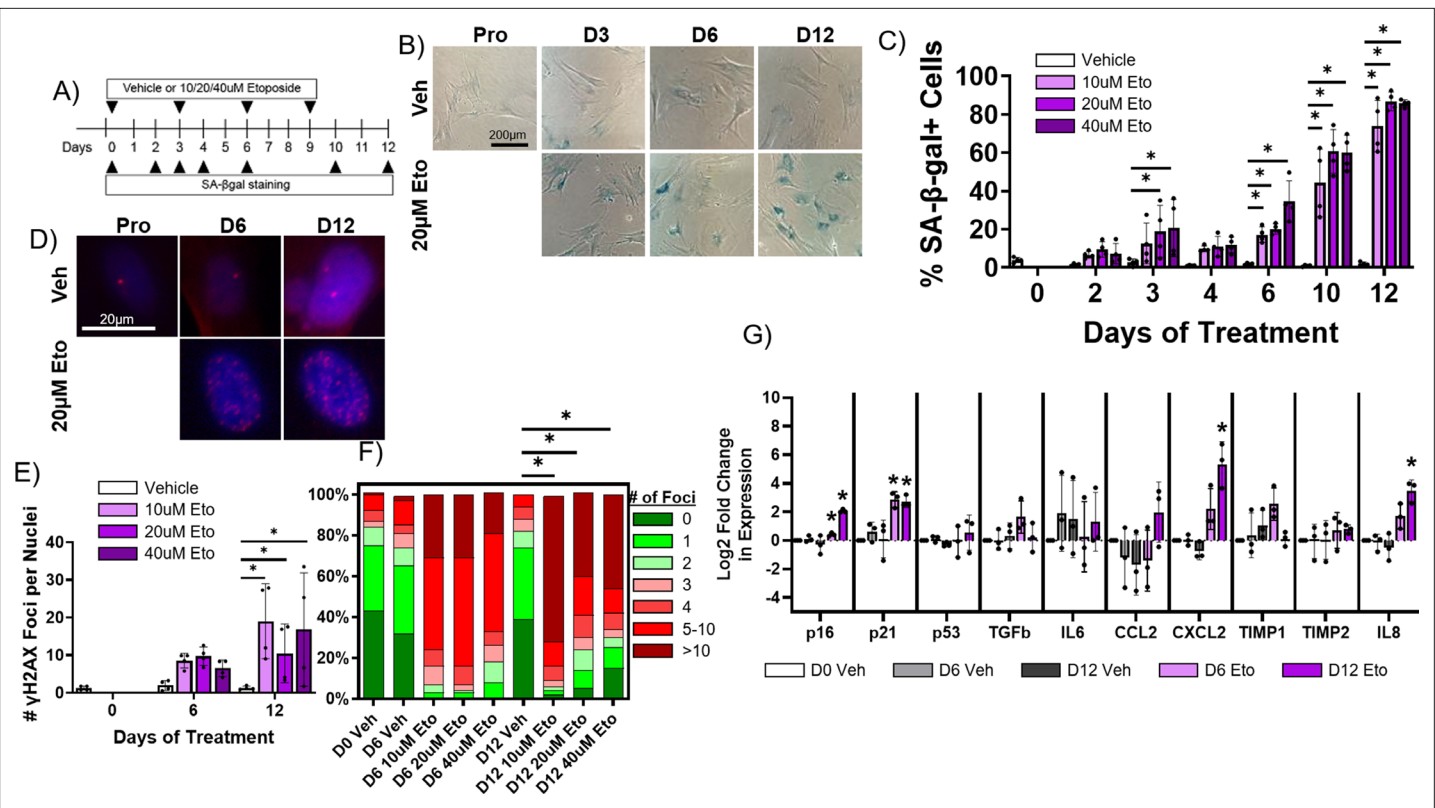

**Figure 6.** Etoposide induces senescence in adult rabbit cardiac fibroblasts. (**A**) Experimental design. (**B**) Representative senescence-associated β-galactosidase (SA-β-gal)-stained images of treated fibroblasts. (**C**) Quantification of % SA-β-gal+ cells from (B). (**D**) Representative nuclear γH2AX-stained images of treated fibroblasts. (**I**) Quantification of # γH2AX foci from (D). (**F**) Quantification of distribution of γH2AX foci from (D). (**G**) Quantification of RT-qPCR of treated fibroblasts. Dots represent average data for experimental replicates, error bars SEM. * p<0.05, one-way ANOVA compared to Vehicle.

The online version of this article includes the following source data and figure supplement(s) for figure 6:

**Source data 1.** Raw data used to create *Figure 6C*.

**Source data 2.** Raw data used to create *Figure 6F*.

**Source data 3.** Raw data used to create *Figure 6G*.

**Figure supplement 1.** Verification of adult rabbit cardiac fibroblast identity.

81.7 ± 18.3%, 59.9 ± 28.1%, and 64.2 ± 42.9% of cells exhibited more than five γH2AX nuclear foci (*Figure 6F*).

Finally, we assessed changes in the expression of senescence and SASP genes in rabbit cardiac fibroblasts treated with vehicle (dimethylsulfoxide [DMSO] alone) or 20 μM etoposide over time. We observed no significant difference in transcript levels of any assayed targets between the untreated, proliferating cells and cells treated with vehicle for 6 or 12 days (*Figure 6G*). For cells treated with 20 μM etoposide for 6 days, we observed a significant increase in the expression of p16 and p21. For cells treated with 20 μM etoposide for 12 days, we observed a significant increase in the expression of p16, p21, CXCL2, and IL-8. Overall, our data indicate that 20 μM etoposide readily induce senescence in the vast majority of rabbit cardiac fibroblasts after 12 days of treatment.

## Senescent adult rabbit cardiac fibroblasts do not appear to affect myocyte electrophysiology in a paracrine manner

Exogenous treatment of individual proinflammatory factors present in the SASP have been shown to induce electrophysiological dysfunction in myocyte in vitro, and several age-associated diseases are exacerbated by chronic inflammatory signaling via the SASP in other tissues (*Francis Stuart et al., 2016*; *Furman et al., 2019*). We hypothesized that SASP factors from senescent cardiac myofibroblasts in the scar or IBZ may act in a paracrine fashion on IBZ myocytes to alter ion channel function,

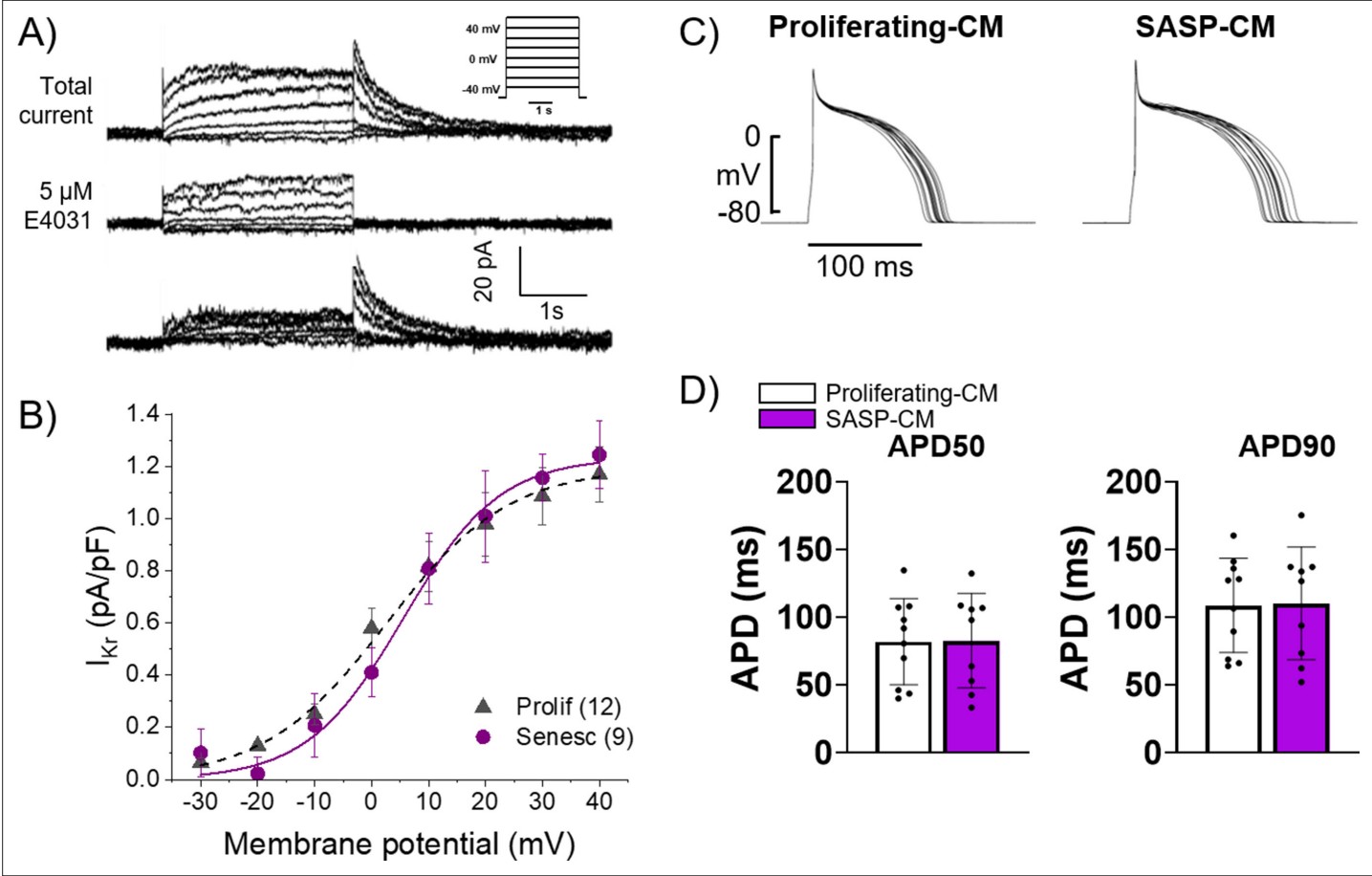

**Figure 7.** Conditioned media from senescent adult rabbit cardiac fibroblasts does not affect action potential duration (APD) or $I_{kr}$. (**A**) Representative current traces showing total current, current after application of 5 µM E4031, and their difference that reveals $I_{kr}$ of primary 3-week-old rabbit myocytes treated with conditioned media from adult rabbit cardiac fibroblast. Peak of the tail of $I_{kr}$ was measured at different potentials. Inset shows voltage protocol. (**B**) Cumulative data of $I_{kr}$ current. Numbers in parenthesis are number of cells analyzed. (**C**) Representative APD traces from primary 3-week-old rabbit myocytes treated with conditioned media from adult rabbit cardiac fibroblasts. (**D**) Quantification of (C). Error bars SEM.

The online version of this article includes the following source data and figure supplement(s) for figure 7:

**Source data 1.** Contains raw data used to create *Figure 7B*.

**Source data 2.** Raw data used to create *Figure 7D*.

**Figure supplement 1.** Twenty-four to 30 hr co-culturing of ventricular cardiomyocytes with senescent or proliferating fibroblasts does not significantly affect action potential duration.

**Figure supplement 1—source data 1.** Raw data used to create *Figure 7—figure supplement 1*.

ultimately prolonging APD and potentially establishing both a substrate and trigger for arrhythmias. To test this hypothesis, we employed an in vitro model using 3-week-old rabbit cardiomyocytes and adult rabbit cardiac myofibroblasts (*Kabakov et al., 2021*). Acutely dissociated cardiomyocytes from 3-week-old rabbits were treated for 30 min with conditioned media collected from adult rabbit cardiac fibroblasts treated for 12 days with either vehicle ('proliferating-CM') or 20 µM etoposide ('SASP-CM'), followed by patch clamping of the myocytes. We measured $I_{kr}$, the main potassium currents regulating repolarization and APD in these cells, in voltage clamp mode. The holding potential was –40 mV. Three seconds depolarizing voltage steps were applied from –30 mV to 40 mV in 10 mV intervals. Experiments were done before and after addition of 5 µM E4031, a specific blocker of $I_{kr}$. The $I_{kr}$ current was determined as E4031 sensitive current (*Figure 7A*). To generate the I-V curve, $I_{kr}$ was quantified by measuring its tail peak right after the end of the depolarizing pulse (*Figure 7B*). We observed no significant difference in $I_{kr}$ in myocytes treated with proliferating-CM versus SASP-CM. We then measured APD in the current-clamp mode in 3-week-old rabbit myocytes treated for 30 min

with either proliferating-CM or SASP-CM. APs were evoked at 0.5 Hz rate, with a 3 ms depolarizing current pulse to about 30% above the threshold of the activation of the AP (*Figure 7C*). For each cell, the average of at least 20 APD traces were used for quantification. We observed no significant difference between myocytes treated with proliferating-CM or SASP-CM in either APD50 or APD90, indicating that the conditioned media had no significant effect on the APD.

Since it is possible that 30 min was too short a treatment of cardiomyocytes with conditioned media, we cultured adult rabbit ventricular cardiomyocytes for 24–30 hr with adult rabbit ventricular cardiac fibroblasts that were either proliferating or induced into senescence. Not only did this experiment extend the duration of cardiomyocyte exposure to SASP factors, but they were acutely produced and not subject to a freeze/thaw that could degrade or alter the SASP factors. Electrophysiological recordings indicated no significant difference in the APD between cardiomyocytes cultures with proliferating versus senescent fibroblasts, or without any fibroblasts at all (*Figure 7—figure supplement 1*). This result provides further evidence that paracrine factors released by senescent fibroblasts likely play a negligible role in the APD prolongation of cardiomyocytes in the rabbit IBZ. We also found no difference in $I_{Kr}$ and $I_{to}$ currents in the fibroblast cardiomyocyte co-cultures compared to cardiomyocytes alone (data not shown) consistent with no significant change in APD due to paracrine SASP factors. Overall, conditioned media from senescent adult rabbit cardiac fibroblasts does not seem to have a paracrine pro-arrhythmic effect on adult rabbit myocytes. In addition, based on the absence of any significant difference in the transient capacitive current kinetics in cardiomyocytes co-cultured with myofibroblasts and cardiomyocytes cultured alone (data not shown), there was no evidence of electrical coupling between the co-cultured cardiomyocytes and myofibroblasts.

## Senescent cardiac fibroblasts can induce pro-arrhythmic changes in myocytes via cell-cell interactions

In addition to paracrine effects, senescent cells can act on nearby cells in juxtacrine fashions to induce senescence and cellular dysfunction, both via signaling through cell surface receptors like NOTCH and via direct transfer of intracellular materials via gap junctions (*Narita, 2019*; *Nelson et al., 2018*; *Nelson et al., 2012*). This is particularly relevant to cardiac rhythm, as cardiac fibroblasts have been shown to directly couple with myocytes and influence their electrophysiology via gap junctions (*Quinn et al., 2016*). Specifically in rabbits, studies have demonstrated functional fibroblast-myocyte gap junction-mediated coupling both via dye transfer assays and via optogenetic assays (*Camelliti et al., 2004b*; *Quinn et al., 2016*; *Kostecki et al., 2021*). To test our hypothesis that senescent myofibroblasts in the IBZ couple with myocytes and promote arrhythmias by altering electrophysiological properties of myocytes, we examined senescence and connexin-43 in frozen cardiac tissue sections from young and aged rabbits 3 weeks post-MI and from adult rabbit cardiac fibroblasts in vitro. First, we quantified the number of senescent cells in the IBZ that were in immediate proximity to myocytes. From SA-β-gal-stained frozen sections, we observed a significant increase in the percent of SA-β-gal+ cells that were adjacent to cardiomyocytes in aged rabbits compared to young (*Figure 8A*). Of cells neighboring myocytes, we also observed a significant increase in the average number of SA-β-gal+ cells per myocyte in aged rabbits compared to young with no difference in the number of SA-β-gal- cells per myocyte (*Figure 8B*). In young rabbits, the average IBZ myocyte had four neighboring non-senescent cells and in aged rabbits, the average IBZ myocyte has four neighboring non-senescent cells and one senescent cell.

To characterize Cx43 expression and localization, we first performed immunofluorescence staining against Cx43 in proliferating and senescent adult rabbit cardiac fibroblasts. In senescent fibroblasts we frequently observed Cx43 lining the interface between adjacent fibroblasts, implying coupling between the cells (*Figure 8C*). We next performed immunofluorescence staining of frozen cardiac sections from young and aged rabbits 3 weeks post-MI against Cx43, γH2AX, and αSMA. Particularly in aged rabbits, we observed αSMA+ cells with three or more γH2AX foci immediately near myocytes at the IBZ with Cx43 appearing at points of contact between the cells, suggesting coupling occurs in vivo (*Figure 8D*, *Figure 8—figure supplement 1*). We observed equivalent results in similarly stained infarcted human ventricular tissue samples, implying the same phenomenon occurs in human cardiac tissue (*Figure 8E*). Importantly, we did not observe evidence of nuclear γH2AX foci in myocytes in the IBZ, consistent with our observations in rabbits. In human tissue, the vast majority of apparently coupled cells were αSMA+, and nearly every coupled γH2AX+ cell was also αSMA+.

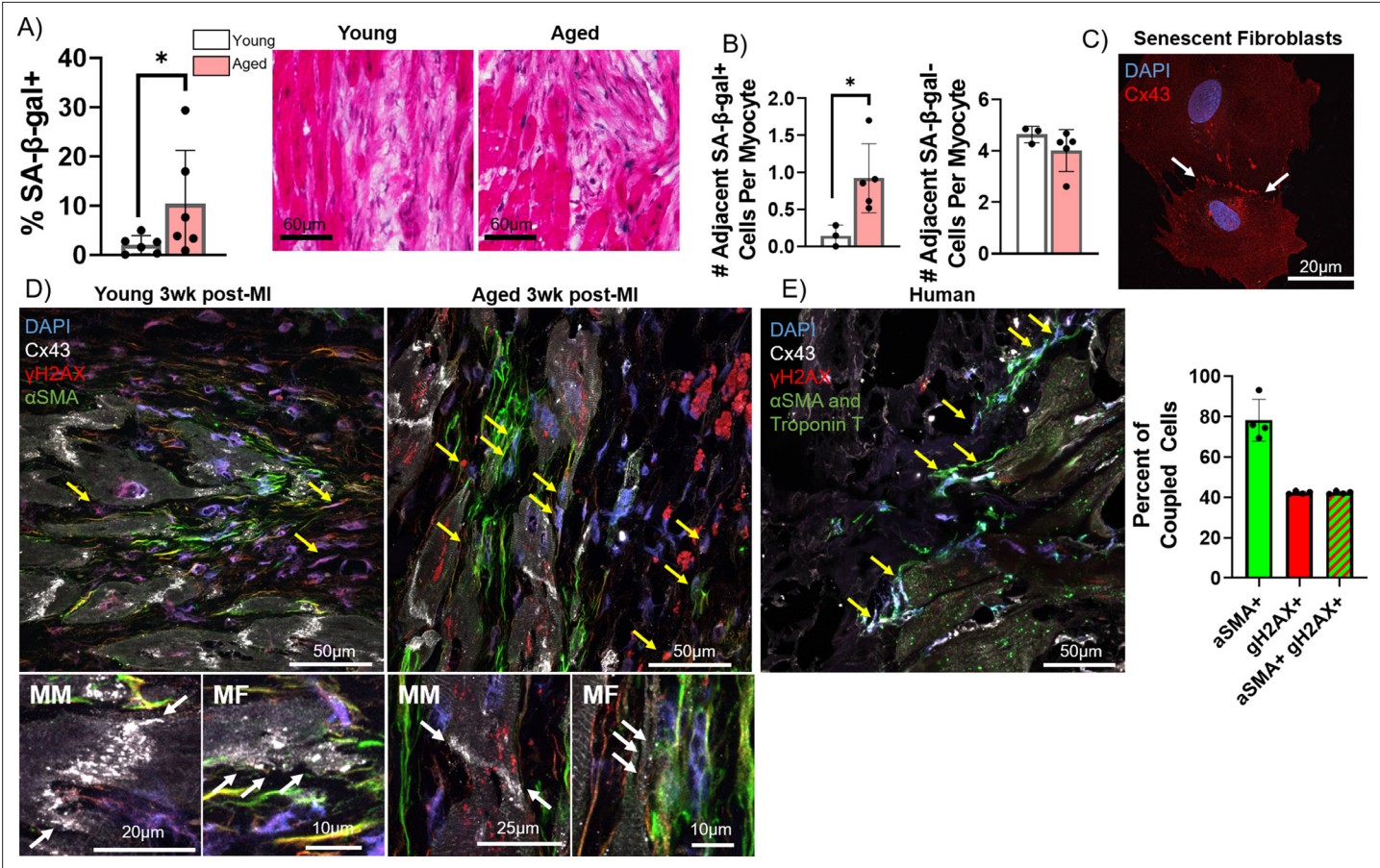

**Figure 8.** Pro-arrhythmic effect of myocytes coupled to senescent fibroblasts compared to non-senescent fibroblasts. (**A**) Quantification of the percent of senescence-associated β-galactosidase (SA-β-gal+) cells adjacent to myocytes (left) and representative SA-β-gal-stained images of the infarct border zone (IBZ) from young and aged rabbits 3 weeks post-MI. (**B**) Quantification of the number of adjacent SA-β-gal+ cells per myocyte (left) and the number of adjacent SA-β-gal- cells per myocyte (right) from the young and aged rabbit IBZ 3 weeks post-MI. (**C**) Representative confocal image of senescent adult rabbit cardiac fibroblast immunofluorescent staining against Cx43. Line of Cx43 between cells indicated with white arrows. (**D**) Representative confocal images of immunofluorescent staining against Cx43, γH2AX, and αSMA of the IBZ of young and aged rabbits 3 weeks post-MI. Yellow arrows indicate nuclei with three or more nuclear γH2AX foci. (**E**) Representative confocal image (left) of immunofluorescent staining against Cx43, γH2AX, αSMA, and troponin T of the IBZ of aged human IBZ samples and quantification of % of coupled cells (right). Yellow arrows indicate nuclei with three or more nuclear γH2AX foci. αSMA and troponin T staining used the same spectral channel secondary antibody (and are therefore both pseudo-colored green) to avoid autofluorescence, as each marker is specific to myofibroblasts and cardiomyocytes, respectively. Dots represent average data for each rabbit, error bars SEM. * p<0.05, two-tailed exact test.

The online version of this article includes the following source data and figure supplement(s) for figure 8:

**Source data 1.** Raw data used to create *Figure 8*.

**Figure supplement 1.** Additional representative images of potential cardiomyocyte-myofibroblast couplings mediated by Cx43.

We next investigated the electrophysiological relevance of potential coupling between senescent cardiac fibroblasts and myocytes by obtaining volumetric and patch clamping data from proliferating and senescent myofibroblasts in vitro and using these data to model the coupling between myocytes and either proliferating or senescent cardiac fibroblasts in silico. We measured the volume of proliferating and senescent cardiac myofibroblasts using confocal microscopy both from frozen sections (*Figure 9A*, left) and from monolayer cultures (*Figure 9A*, right) using immunostaining against the lectin wheat germ agglutinin (to delineate the cell surface). In both cases, we observed that senescent myofibroblasts had a significantly larger volume compared to non-senescent ones, consistent with previous literature, and an overall increase in the size of in vitro myofibroblasts compared to those from tissue sections (*Childs et al., 2019*). We next characterized the electrophysiology of single proliferating or senescent adult rabbit cardiac myofibroblasts in vitro via whole-cell patch clamping.

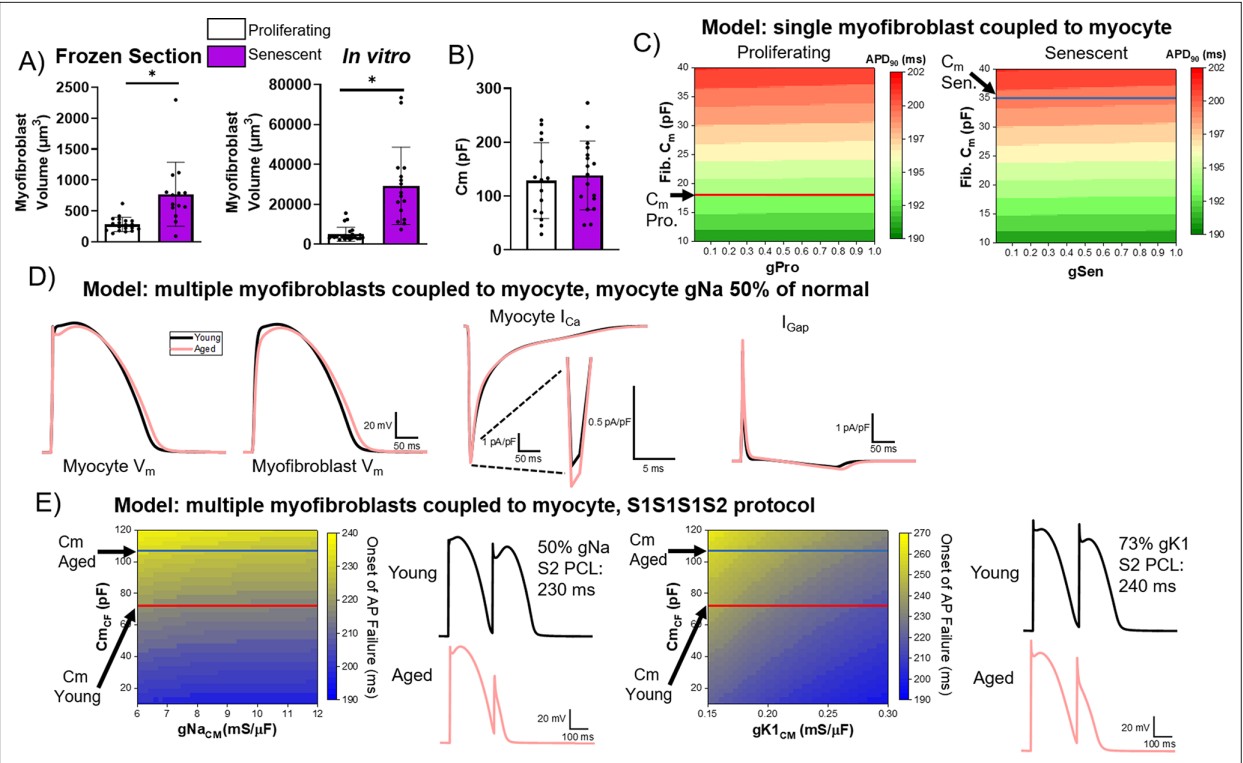

**Figure 9.** Computational modeling of myocyte-myofibroblast interactions. (**A**) Quantification of proliferating and senescent myofibroblast volume from immunofluorescent staining against WGA and γH2AX of frozen tissue sections (left) and adult rabbit cardiac fibroblasts in vitro. (**B**) Quantification of capacitance of patch clamped proliferating and senescent adult rabbit cardiac myofibroblasts. (**C**) Modeling of action potential duration (APD)90 from a single myocyte coupled to a single proliferating or senescent fibroblast with gradients of myofibroblast conductance (gPro and gSen) and myofibroblast capacitance. (**D**) Modeling of myocyte (first panel) and myofibroblast (second panel) action potential traces, as well as $I_{Ca}$ and $I_{Gap}$ traces from young and aged modeling conditions. (**E**) S1S2 modeling of young and aged conditions showing onset of action potential failure with gradients of Na channel conductance (gNa) and myofibroblast capacitance, with representative action potential traces (left) and onset of action potential failure with gradients of myocyte inward rectifier channel conductance (gK1) and myofibroblast capacitance, with representative action potential traces (right). Dots represent average data for each cell, error bars SEM. * $p<0.05$, two-tailed exact test.

The online version of this article includes the following source data and figure supplement(s) for figure 9:

**Source data 1.** Raw data used to create *Figure 9*.

**Figure supplement 1.** Additional parameters for myocyte-myofibroblast modeling.

The capacitance of proliferating and senescent cardiac fibroblasts in vitro was 128.7±70.3 pF for proliferating cells and 138.4±64.0 pF for senescent cells, and along with the volumetric measurements, we extrapolated the capacitance for proliferating and senescent rabbit cardiac fibroblasts in vivo to be 18 pF and 35 pF (see Methods), respectively (*Figure 9B*). Patch clamping of proliferating and senescent myofibroblasts revealed senescent myofibroblasts to have a slight but nonsignificant increase in peak current compared to proliferating myofibroblasts (*Figure 9—figure supplement 1*). The voltage-current relationship for each type of cell was fitted with a sigmoidal Boltzmann equation to model the myofibroblast currents.

Using these in vitro measurements, we modeled coupling interactions between single myocytes and either a single proliferating or senescent myofibroblast with a gradient of myofibroblast conductance and capacitance values. Our modeling showed a slight decrease in APD90 with increasing conductance values, and a strong increase in APD90 with increasing capacitance (*Figure 9C*). With the extrapolated capacitance values for proliferating and senescent myofibroblasts, our modeling suggests that myocyte coupling to senescent myofibroblasts results in prolongation of APD. Additionally, we simulated interactions in young tissue (i.e. one myocyte interacting with four non-senescent myofibroblasts) and in aged tissue (i.e. one myocyte interacting with one senescent myofibroblast and four proliferating myofibroblasts), adjusting Na+ channel conductance to 50% of normal values to account for ion channel remodeling observed at the IBZ in canine MI (*Hegyi et al., 2018*). Simulating

myocyte APs from young and aged tissue (*Figure 9D*, first panel), our modeling showed that the aged condition resulted in a more prominent notch at phase 1 as well as a prolongation of the APD compared to young (*Figure 9D*, second panel), consistent with our optical mapping observations of aged rabbits. Our modeling showed that the main cause of APD prolongation in the aged condition is due to the lower plateau $V_m$ leading to a larger $I_{Ca}$ (–7.4 pA/pF in the aged vs. –7.2 pA/pF in the young condition) (*Figure 9D*, third panel). The gap junctional currents between myocyte and fibroblasts were greater in the aged condition, 8.1 pA/pF vs. 7.4 pA/pF in the young condition (*Figure 9D*, fourth panel), indicating that myocyte-fibroblast coupling causes the lower plateau $V_m$ in the aged condition to increase $I_{Ca}$ driving force thereby prolonging APD in the aged condition.

To further investigate potential arrhythmogenic juxtacrine mechanisms of senescent myofibroblasts in silico, we modeled S1S2 protocol in young and aged conditions and investigated at which point premature stimulation of S2 beat fails to induce AP. Modeling the CL of the S2 beat at which AP failure occurred at variable values for myocyte Na channel conductance (gNa, normal = 12 mS/μF) and fibroblast capacitance ($Cm_{CF}$) revealed a slight decrease in the CL at onset of AP failure with increasing gNa values and an increase in the CL at onset of AP failure with increasing $Cm_{CF}$ (*Figure 9E*, left). At our extrapolated capacitance values for the young and aged conditions, AP failure occurs at greater CL values under aged conditions compared to young. Representative modeled AP traces for young and aged simulations during S1S2 protocol are shown in *Figure 9E*, left. We also modeled the occurrence of AP failure with varying myocyte inward rectifier channel conductance (gK1, normal ~0.30 mS/μF) values and myofibroblast capacitance values (*Figure 9E*, right). Compared to our modeling of various gNa values, we observed a steeper relation of increasing gK1 values to decreasing CL values at onset of AP failure for a given capacitance value. Similarly, at the extrapolated values for young and aged capacitance, we found onset of AP failure occurred at higher CL values under aged conditions compared to young when accounting for ~73% of normal gK1 as experimentally observed in the pig IBZ (*Hegyi et al., 2018*). These results support that coupling with senescence fibroblasts increases risks for conduction failure, heightening arrhythmogenicity in the aged MI condition.

## Discussion

Although the age-associated accumulation of senescent cells and the related detriments to cardiac function and chronic inflammation have been previously described (*Walaszczyk et al., 2019*), the relationship between senescence arising during and after scar formation post-MI and arrhythmogenesis is not well understood. Our experiments show that accumulation of senescent myofibroblasts in the scar is associated with a tissue microenvironment more susceptible to cardiac arrhythmias and elevated inflammatory signaling. Additionally, our findings suggest that senescent myofibroblasts in the IBZ may be exerting pro-arrhythmic effects on surviving myocytes in a cell-cell interaction mediated by myocyte-fibroblast coupling via gap junctions. We propose that persistence of senescent myofibroblasts is a major driver of arrhythmogenic tissue remodeling with age post-MI, and that the timely pharmacological elimination of senescent cells might alleviate risk of arrhythmias particularly in elderly infarcted patients.

From our MI surgical procedures, we observed significantly increased mortality of aged rabbits mostly due to peri-procedural VT/VF (*Figure 1*). These results are consistent with the age-associated increase in out-of-hospital mortality from cardiac arrhythmias resulting from MI (*Benjamin et al., 2019*). The reason for this increase in peri-procedural mortality with age is likely related to our previous observations of increased interstitial fibrosis, disorganization of the His-Purkinje system, and altered myocyte $Ca^{2+}$ handling with age in non-infarcted aged rabbits, in that these age-associated changes likely increase peri-procedural mortality by lowering the threshold for arrhythmias triggered by the acute ischemic insult (*Cooper et al., 2012*; *Murphy et al., 2019*; *Terentyev et al., 2014*). Based on our previous observations showing low levels of senescence in non-infarcted young and aged rabbits (not shown), we posit that senescence likely does not play a role in peri-procedural arrhythmias.

The anatomical characteristics of the scar, including scar size, functional IBZ area, and dense and interstitial fibrosis, could influence age-associated differences in arrhythmogenesis. However, the fact that we did not observe any significant difference in the timecourse of scar size, IBZ area, and overall or interstitial fibrosis suggests that age-associated changes in other cellular processes, including cellular senescence, are driving the increase arrhythmogenicity that we observe from optical mapping studies. Our findings of overall scar size are consistent with our previous study showing 30% infarcted

area of the left ventricle from our MI surgical procedure, and the present data suggest no epicardial infarct expansion occurs over time in young or aged rabbits. (*Morrissey et al., 2017*).

Electrophysiological remodeling at the IBZ includes separation of myocyte bundles via interstitial and replacement fibrosis and significant disruptions of APD and reductions in conduction velocity, myocyte-myocyte coupling, and anisotropy ultimately permitting unidirectional conduction block, sustained reentry, and subsequent VT/VF (*Kolettis, 2013*; *Rutherford et al., 2012*; *Di Diego and Antzelevitch, 2011*). Although these changes are well characterized in young animal models, we sought to investigate age-associated changes in these processes and their underlying mechanisms. From our optical mapping studies, we observed aged rabbit hearts 3 weeks post-MI exhibited more extreme arrhythmogenic remodeling at the IBZ compared to young, including relative prolongation of APD as compared to the RZ, spatially discordant alternans near the IBZ at relatively slow heart rate, and higher frequency of sustained spontaneous VF following burst pacing. These pathological characteristics indicate that aged rabbits undergo remodeling at the IBZ differently than young animals, culminating in a stronger substrate for arrhythmias anchored at the IBZ. Specifically, prolonged APD near the IBZ has been shown to facilitate a reentrant circuit around the scar (*Campos et al., 2019*). Meanwhile, in young rabbits, the significantly shorter APD at the IBZ compared to the RZ and scar can also induce and maintain arrhythmia through increasing APD dispersion and AP conduction block. This finding of shortened APD in the border zone of young rabbits is consistent with previous data using relatively young Yucatan minipigs (*Hegyi et al., 2018*). However, young infarcted rabbits experience less arrhythmias than aged infarcted rabbits, as indicated by the lack of observed voltage alternans at slow heart rate and slower VF frequency. The observation of similarly slower conduction velocity at the IBZ relative to the RZ in both young and aged rabbits is consistent with our histological observations showing no difference in the number or length of IBZ protrusions (*Figure 2B*).

The accumulation of senescent cells in many tissues has been shown to exacerbate age-associated pathologies via paracrine or juxtacrine mechanisms. Studies in mice have demonstrated increased senescence in the scar post-MI with age, the pharmacological clearance of which limits scar size and improves cardiac function and outcome (*Walaszczyk et al., 2019*). However, little is known about whether these findings translate to large animal models with more electrophysiological relevance to the human heart, or the mechanisms through which senescent cells might promote arrhythmogenesis in the aged infarcted heart. Here, we show that cellular senescence as measured by SA-β-gal and γH2AX staining between young and aged rabbits is similar for the first 2 weeks post-MI but persists in aged rabbits at 3 and up to 12 weeks post-MI in the scar (*Figure 4A–E*) and is associated with higher local expression of inflammatory SASP factors (*Figure 4H*). Consistent with previous findings from our group and others (*Zhu et al., 2013*), we found very few SA-β-gal-positive cells in the RZ of young or aged rabbits. These data, along with our previous results assessing young and aged non-infarcted rabbits, suggest a low baseline of senescence in the normal young and aged rabbit heart which is reasonable considering the relatively low proliferation of myocardial cells under baseline conditions. The similarly low levels of SA-β-gal in young and aged rabbits 2 weeks post-sham infarction indicate that the age-associated significant difference in senescence post-MI was not an artifact of the MI procedure. Additionally, we did not observe any evidence of senescent cardiomyocytes in young or aged rabbits at any timepoint (*Figure 4C*, bottom) as well as from human ventricular tissues samples from SA-β-gal staining, which differs from previous reports of mouse tissue (*Anderson et al., 2019*). This important observation suggests that the rabbit heart could be a more appropriate animal model than the mouse to study the effects of cardiac aging relevant to humans. Interestingly, we observed no difference in either total fibrosis of the LV free wall or interstitial fibrosis over time between young and aged rabbits even though we observed persistence of senescence in the aged rabbit heart over time compared to young. It is well known that the content of the SASP changes over time and depending on cell type and the induction source, and so it is possible that pro-fibrotic and anti-fibrotic SASP factors are present in functionally equivalent amounts ultimately resulting in no difference in fibrosis. This hypothesis is supported by our RT-qPCR data, indicating upregulation in expression of the pro-fibrotic factors TGF-β, TIMP1, and TIMP2 as well as anti-fibrotic factor MMP9. Further investigation into this hypothesis is warranted.

From our immunofluorescence staining, the high degree of αSMA signal in young and aged rabbits at 1 and 2 weeks post-MI and the subsequent decrease in αSMA signal is consistent with genetic lineage tracing studies in mice which suggest that myofibroblasts remain present in the scar but lose

αSMA expression (*Fu et al., 2018*). Our observation that the percent of γH2AX+ cells that are αSMA+ remains high at 2 weeks post-MI when αSMA signal begins to decline suggests that most of the senescent cells present in the scar are αSMA+ myofibroblasts that lose αSMA expression after 2 weeks post-MI. This observation is consistent with previous studies characterizing the identity of senescent cells in the scar post-MI in mice (*Zhu et al., 2013*; *Meyer et al., 2016*). The relatively lower percent of CD31+ cells at any timepoint between young and aged rabbits is consistent with previous reports of cellular composition in the scar (*Pinto et al., 2016*). The small percent of γH2AX+ foci that were CD31+ reinforces our conclusion that myofibroblasts are the majority of cells undergoing senescence in the young and aged heart post-MI.

Our RT-qPCR data of young and aged rabbits 3 weeks post-MI further demonstrates differences between the senescence profiles of the aged rabbit and the aged mouse. Consistent with our SA-β-gal and γH2AX data, we observed an increase in the expression of p16, p21, and p53 in aged rabbits compared to young. However, the only SASP factors for which we measured a significant change in expression in the scar zone between young and aged were CXCL2 and TIMP2, whereas the contents of the cardiac SASP were largely different between young and aged mice (albeit with a non-canonical SASP) (*Anderson et al., 2019*). Given these results, special attention should be placed on conclusions for the relevance of senescence studies performed in mice in regard to their clinical relevance for humans. Similar to our SA-β-gal data, we observed a similar but lower magnitude of changes in overall expression between the young and aged rabbit IBZ, and no differences in senescence and SASP gene expression between the young and aged RZ. The significant decrease in IL-6 expression in the aged IBZ compared to young IBZ is interesting, however senescence and SASP signatures are known to be variable depending on cell line and senescence induction method, and this decrease in IL-6 expression has also been observed in studies of young and aged non-infarcted murine myocytes (*Anderson et al., 2019*). Overall, our data suggest that senescence and the inflammatory SASP is largely confined to the scar.

To investigate the clinical relevance of our observations of persistent senescence in the aged infarcted rabbit heart, we measured senescence in aged infarcted human ventricular frozen sections. We observed a relatively high percent of cells in the scar exhibited nuclei with three or more γH2AX foci (*Figure 5C*), indicating the presence of senescent cells in the human tissue, which may contribute to arrhythmogenic tissue remodeling leading to increased incidence of potentially lethal arrhythmias in the chronic phase post-MI. These γH2AX senescence data are similar to what we observed in the aged rabbit 12 weeks post-MI. From our αSMA analysis of human tissue, we observed a high percent of cells were αSMA+ unlike the progressive decrease in αSMA+ cells over time that we observed in the young and aged rabbits. Because the specific timing of MI is difficult to diagnose especially out-of-hospital, we are unable to ascertain how long post-MI the samples were at time of death. If senescent cells indeed persist in the aged human scar as we hypothesize, it is possible that growth factors like TGF-β in the chronic SASP activate fibroblasts in the months-to-years post-MI. Alternatively, the content of the SASP could differ from what we observe in rabbits, contributing to differential activation of myofibroblasts. Similar to our observations in young and aged rabbits, our γH2AX and αSMA double immunofluorescence assays indicate that the majority of cells with three or more γH2AX nuclear foci were also αSMA+, further suggesting that myofibroblasts are the predominant cell type undergoing senescence. Relatively fewer cells with three or more γH2AX nuclear foci were either vWF+ or CD68+, consistent with previous reports in mice, suggesting relatively fewer senescent cells are endothelial cells or macrophages, respectively (*Zhu et al., 2013*).

To better study the mechanisms by which senescence might be promoting arrhythmias with age post-MI, we established an in vitro method to induce senescence in adult rabbit cardiac fibroblasts. Low doses of the drug etoposide induced senescence, measured by SA-β-gal and γH2AX staining as well as RT-qPCR in a time-depending manner, with the vast majority of cells positive for senescence markers after 12 days of treatment with 20 μM etoposide (*Figure 6*). These data establish a platform to future studies screening the efficacy of senolytic drugs to eliminate senescent cells post-MI.

To investigate whether senescent cells might have a paracrine effect on myocytes, we treated primary isolated myocytes from 3-week-old rabbits with conditioned media harvested from proliferating (proliferating-CM) or senescent (SASP-CM) fibroblasts (*Figure 7*). From our patch clamping studies, we observed no significant difference in $I_{Kr}$ current or in APD between proliferating-CM or SASP-CM groups. Separately, we co-cultured isolated myocytes with proliferating or senescent cardiac

fibroblasts for longer durations and also found no significant difference in $I_{Kr}$ or APD. These findings suggest that senescent rabbit fibroblasts likely do not act in a paracrine manner to disrupt the electrophysiology of rabbit myocytes, but further study is needed to verify this. Because we are limited by the number of commercially available assays to determine the specific contents of the rabbit SASP, we are unable to exclude the possibility that the conditioned media lacked physiologically relevant levels of SASP factors to induce a significant effect. Besides, components in the media used or the absence thereof might have blunted a paracrine effect. One could also envisage that degradation of conditioned media occurred between harvest and treatment, as well as potential issues with incubation period and responsiveness of the 3-week-old myocytes to the SASP. Additionally, transfer of the cells from the conditioned media to the patch clamping media might have allowed for rapid recovery of the cardiomyocytes prior to patch clamp recording, which would not be detectable from the data presented.

We next investigated whether senescent fibroblasts might directly couple with myocytes at the IBZ to influence pathological electrophysiological remodeling. Homo- and hetero-typic coupling occurs in the healthy and diseased heart via connexins, including Cx43 (*Kohl and Gourdie, 2014*). We observed a significantly higher percentage of senescent cells neighboring IBZ myocytes in aged rabbits 3 weeks post-MI compared to young. Our immunofluorescence studies suggest that Cx43 facilitates homocellular coupling between senescent adult rabbit cardiac fibroblasts, as well as heterocellular coupling between senescent myofibroblasts and myocytes at the IBZ of aged rabbits and human frozen sections.

Our computational modeling of proliferating and senescent myofibroblasts as well as simulations of cardiomyocyte-myofibroblast interactions in young and aged tissue shed critical light for explaining mechanisms by which senescent myofibroblasts might promote arrhythmogenic tissue remodeling with age post-MI. Although electrical remodeling in the IBZ is a complex phenomenon with multiple contributing factors, our results indicate that myocyte-fibroblast interaction becomes a potential contributor to arrhythmogenesis in the aged MI condition. A large, flattened morphology is one of the hallmark characteristics of senescent cells in other models (*Kuilman et al., 2010*), and we observed this for cardiac myofibroblasts both in vitro and in vivo. The larger size of senescent myofibroblasts likely explains the greater extrapolated values for in vivo capacitance, and all differences in modeling between proliferating and senescent myofibroblasts are driven by the larger capacitance of senescent myofibroblasts. Importantly, the prolongation in APD from our modeling of aged condition relative to young is consistent with our optical mapping findings from young and aged rabbits 3 weeks post-MI and validates the relevance of our modeling. From our S1S2 modeling under young and aged conditions, we would predict that aged rabbits would have higher risks for conduction block than young rabbits. Although we did not observe a significant difference in the CL at which VF was induced, VF frequencies were higher and heterogeneous in the IBZ (*Figure 3D*), suggesting that the prolongation of APD in myocytes coupled to senescent myofibroblasts would result in lower threshold for conduction block and wavebreak, overall driving faster VF frequency. These observations are consistent with our hypothesis that myocyte coupling to senescent myofibroblasts in the IBZ of aged rabbits leads to pro-arrhythmic electrophysiological remodeling and increases risk of reentrant current anchored to the IBZ.

In summary, we have characterized an age-associated accumulation of senescent myofibroblasts in the aged infarcted rabbit heart, and our findings suggest that these senescent cells remodel the electrophysiology of IBZ myocytes at least via direct myocyte-fibroblast coupling mediated by Cx43, which would increase risk of cardiac arrhythmias. Previous studies have shown that pharmacological elimination of senescent cells with age via the senolytic drug navitoclax limits cardiac fibrosis in infarcted and non-infarcted mice and improves the survival and cardiac function of infarcted young and aged mice, but little is known about the effects of senescence on arrhythmogenesis with age (*Walaszczyk et al., 2019*; *Anderson et al., 2019*). Our findings demonstrate an important role of persistent senescence in arrhythmogenesis with age post-MI, suggesting the use of senolytic therapy to mitigate senescence burden with age might be promising strategy to mitigate chronic arrhythmias in aged infarcted individuals.

## Methods

### Animal ethical statement

This investigation conformed with the current Guide for Care and Use of Laboratory Animals published by the National Institutes of Health (NIH Publication, Revised 2011) as well as the standards recently delineated by the American Physiological Society ('Guiding Principles for Research Involving Animals and Human Beings') and was approved by the Institutional Animal Care and Use Committee of Rhode Island Hospital (Permits numbers 5001-21 and 5040-22). Young (≤1 year of age) and aged (≥4 years of age) female New Zealand White rabbits were infarcted as previously described (*Morrissey et al., 2017*; *Ziv et al., 2012*). Rabbits received either a peri-procedural 5 mg/kg/hr amiodarone infusion followed by a single 400 mg oral dose of amiodarone 1 day post-MI, or a peri-procedural 1 mg/kg/hr amiodarone infusion followed by 120 µg/kg/hr IV amiodarone continually for 3 days post-MI delivered by iPRECIO pump (Data Sciences International, St. Paul, MN, USA) implanted into the right jugular vein.

### Tissues

Rabbits were anesthetized with ketamine (60 mg/kg), xylazine (15 mg/kg IM), buprenorphine (0.03 mg/kg SQ), and sodium pentobarbital (150 mg/kg IV). Hearts were excised as previously described (*Morrissey et al., 2017*) and washed in PBS. Gross anatomy photos of the intact LV were taken immediately after beating heart harvest to assess percent scar size of the LV free wall in Adobe Photoshop by freehand selection of the scar and the LV free wall; analysis for each heart was performed by three blinded individuals. Heart tissue for histological analysis was embedded in OCT compound solution (Fisher Scientific, Waltham, MA, USA) and rapidly frozen on liquid nitrogen. Heart tissue for biochemical analysis was snap-frozen in liquid nitrogen and stored immediately at –80°C. Cryosectioning was performed at 6 µm thickness using a Leica CM1950, and slides were immediately fixed in cold 4% PFA in PBS for 15 min followed by a wash in PBS for 15 min, with the exception of slides with Cx43 staining, in which slides were fixed in cold acetone for 10 min.

Interdigitation analysis was performed on cryosections stained with H&E as previously described (*Fischer et al., 2008*) and imaged using an Aperio ScanScope (Leica Biosystems, Buffalo Grove, IL, USA) and analyzed using Aperio ImageScope v12 software (Leica Biosystems, Buffalo Grove, IL, USA). The freehand pen tool was used to mark a continuous IBZ, and the ruler tool was used to determine the number and length of interdigitations protruding from the continuous IBZ marker into the myocardia. All interdigitation analysis was performed by at least three blinded individuals, and multiple cryosections per heart were analyzed.

### Optical mapping of epicardial surface and data analysis

Young and aged hearts were prepared as described previously (*Liu et al., 2018*; *Kim et al., 2019*). Hearts were stained with the voltage-sensitive dye di-4-ANEPPS, and ECG and perfusion pressure were constantly monitored (PowerLab, ADInstruments, Sydney, Australia). A total of 5 young and 18 aged rabbits were used. The imaging apparatus has been previously described (*Choi et al., 2007*). The data were analyzed with a custom-built software program developed in Interactive Data Language [Harris Geospatial Solutions, publicly available at https://github.com/arvinsoepriatna/AP_Analysis_Routines_Cardiotoxicity_Microtissues copy archived at swh:1:rev:3172027cbd78883c5c6e464524af-ea6fa213cfad (*Soepriatna, 2023*) under General Public License]. APDs and APD maps were calculated via the difference between activation and repolarization timepoints, determined from fluorescence (F) signals by calculating $(dF/dt)_{max}$ and $(d^2F/dt^2)_{max}$. Data were filtered using a temporal polynomial filter (third order, 13 points) (*Kim et al., 2015*). AP conduction velocity was calculated using the spatial gradient of activation time (11×11 pixels) and spatial resolution of acquired image (200 µm/pixel).

### Stimulation protocol

The atrioventricular node was ablated to control heart rate using a cautery unit (World Precision Instruments, Sarasota, FL, USA). After ablation, the rabbits exhibited a slower heart rate, with the cycle length typically ranging from 700 ms to 2000 ms. If the cycle length was longer than 2000 ms, the base of the right ventricle was paced at a cycle length of 2000 ms. Ramp pacing protocol: Hearts were paced at CLs that were successively decreased by 10 ms until loss of 1:1 capture or induction of VF or VT (*Brunner et al., 2008*; *Banville and Gray, 2002*; *Hayashi et al., 2007*).

## SA-β-gal assay

### Frozen tissue

SA-β-gal staining was performed with modifications as previously described (*Dimri et al., 1995*; *Debacq-Chainiaux et al., 2009*). Tissue sections immediately after cryosectioning were immersed in 0.5% glutaraldehyde (in 1× PBS) for 15 min then placed in fresh 1× PBS. Sections were then incubated at 37°C with no $CO_2$ with fresh β-galactosidase staining solution (40 mM citric acid/Na phosphate buffer, 5 mM $K_4[Fe(CN)_6]$·$3H_2O$, 5 mM $K_3[Fe(CN)_6]$, 150 mM sodium chloride, 2 mM magnesium chloride, and 1 mg·ml−1×-gal) for 16 hr. Sections were counterstained with H&E, and slides were imaged using an Aperio ScanScope (Leica Biosystems, Buffalo Grove, IL, USA) with a 40× Objective. Images were analyzed using Aperio ImageScope v12 software (Leica Biosystems, Buffalo Grove, IL, USA). The percent of SA-β-gal-positive cells was calculated by hand counting the number of SA-β-gal-positive and SA-β-gal-negative cells from at least five randomly chosen images, using the counter tool in ImageJ, by at least three individuals.

## Cell culture measurements

SA-β-gal staining of rabbit cardiac fibroblasts in cell culture plates was performed as previously described, in triplicate (*Bandyopadhyay et al., 2005*). Cells were imaged in PBS at using a Nikon Eclipse TE2000U with a Retiga EXI color camera. Images were acquired with a 20× FLUOTAR NA Objective under differential interference contrast. At least five images were acquired per biological replicate. The percentage of SA-β-gal-positive cells were calculated by hand in ImageJ as described above.

## Immunofluorescence staining

### Frozen tissue

Frozen sections were stained via immunofluorescence with some modifications as previously described (*Maity et al., 2013*). Briefly, 6-µm-thick frozen sections were acquired and fixed as described above. Tissue was encircled with a PAP pen. For intracellular targets, frozen sections were permeabilized with 0.1% Triton X-100 in PBS for 30 min. Slides were then washed with PBS and blocked with 3% bovine serum albumin, normal donkey serum, or normal goat serum in PBS for 1 hr. Slides were then incubated with primary antibody diluted in blocking buffer overnight at 4°C in the dark, then washed in PBS, and incubated in secondary antibody in blocking buffer for 2 hr in the dark. Slides were then mounted with Prolong Gold Reagent with DAPI and imaged with a Nikon Ti2 confocal microscope with an A1R scanner. At least five images with at least 10 Z-stacks were acquired randomly for each sample (images analyzed were approximately 200 µm width, 200 µm height, and 6 µm depth). Antibodies used are listed in *Supplementary file 1*.

The number and percentage of cells with γH2AX and αSMA signal were derived from maximum projection images by hand using the counter tool in ImageJ. All measurements were performed by two blinded individuals. A γH2AX nuclear focus was identified if at least one distinct focus overlapped the DAPI-positive nucleus; large, continuous regions of signal in the γH2AX channel inside a nucleus and any signal outside the nucleus were considered artifact and were excluded from analysis. A cell was considered αSMA-positive if signal in the αSMA channel was present within 10 µm of a nucleus and displayed the expected morphology.

## Cell culture

Cells in triplicate were fixed in wells with 4% PFA for 15 min, then washed with PBS. Cells were incubated with blocking solution (5% normal serum corresponding to host of secondary antibody, 0.3% Triton X-100 in 1× PBS) for 1 hr. Cells were then treated with primary antibody diluted in blocking solution for 16 hr at 4°C, then washed in PBS. Cells were then incubated in secondary antibody diluted in blocking solution for 2 hr in the dark, washed in PBS, then mounted with Prolong Gold Reagent DAPI and imaged with a Nikon Eclipse TE2000U inverted microscope at 40×. Multiple Z-stack images with 0.5 µm step length were taken per view, and at least five views per replicate were imaged. To computationally calculate the number of γH2AX foci per nucleus, a maximum intensity projection was created for each Z-stacked image and individual nuclei were cropped in ImageJ, and a custom analysis macro was run.

## RT-qPCR
### Tissue

Total RNA was harvested from 40 mg to 50 mg of LV tissue (scar zone, IBZ, and RZ) using Trizol reagent (Invitrogen). Nucleic acid concentration of the RNA was determined using a Nano-drop 2000c Spectrophotometer (Thermo Fisher, Waltham, MA, USA). Reverse transcription was performed using the iScript reverse transcription kit (Bio-Rad, Hercules, CA, USA) and a C1000 Thermocycler (Bio-Rad, Hercules, CA, USA). Primer sets were designed for each gene of interest as well as endogenous control. National Center for Biotechnology Information gene database (https://www.ncbi.nlm.nih.gov/gene) was used to obtain accession numbers for each gene, and Primer Blast tool (https://www.ncbi.nlm.nih.gov/tools/primer-blast/) was used to yield several primer sets for each gene. Primer sets were selected for each gene of interest for subsequent testing. The efficiencies of the selected primer set for each gene were determined (*Supplementary file 2*) to accurately calculate normalized change gene expression (ΔCT). Real-time qPCR was performed using SYBR Green chemistry (Bio-Rad) and a ViiA 7 Real Time System (Thermo Fisher, Waltham, MA, USA). Primer efficiency was calculated using 40 ng, 4 ng, 0.4 ng, 0.04 ng, and 0.004 ng of cDNA from rabbit LV free wall tissue performed in triplicate for each primer set. SRP14 internal control (*Pilbrow et al., 2008*) and no template were used as positive and negative controls, respectively. Standard curves, melt curves, and primer efficiency were reported by the ViiA7 QuantStudio software.

## RT-qPCR
### In vitro

Cells plated in 10 cm plastic uncoated dishes (CytoOne, USA Scientific) were harvested with Trizol reagent (Invitrogen). Nucleic acid concentration of the RNA was determined using a Nanodrop 2000c Spectrophotometer (Thermo Fisher, Waltham, MA, USA). Reverse transcription was performed using the iScript reverse transcription kit (Bio-Rad, Hercules, CA, USA) and a C1000 Thermocycler (Bio-Rad, Hercules, CA, USA). Real-time qPCR was performed using SYBR Green (Bio-Rad) and a ViiA 7 Real Time System (Thermo Fisher, Waltham, MA, USA) as specified by the manufacturer. Primers used are listed in *Supplementary file 2*. For each condition, senescence of cells plated and treated in parallel was confirmed visually via SA-β-gal staining: all vehicle-treated cells showed low percent of SA-β-gal+ cells and all etoposide-treated cells showed a high degree of SA-β-gal+ cells.

## Adult rabbit cardiac fibroblast isolation and culturing

Primary adult rabbit cardiac fibroblasts (rabbit cardiac fibroblasts) were isolated from aged (>4 years) New Zealand White rabbits with modifications from previously described (*Cooper et al., 2013*), by Langendorff perfusion with a solution containing collagenase II (#CLS-2, Worthington Biochemical, Lakewood, NJ, USA). After isolation, instead of using the myocyte-containing pellet, the supernatant was centrifuged three times at 400 rpm for 5 min, transferring the supernatant to a new tube each time. After the third centrifugation, the supernatant was aspirated and the pellet was resuspended in rabbit fibroblast media (DMEM/F-12 [Life Technologies] supplemented with 10% fetal bovine serum, penicillin [100 µg/mL], and streptomycin [0.1 µg/mL] [#P4333, Sigma, St. Louis, MO, USA]). Cells were plated onto 10 cm plastic dishes (CytoOne, USA Scientific) and grown for three passages in a 37°C incubator at atmospheric oxygen before cell identity immunofluorescence staining (*Figure 5—figure supplement 1*) and subsequent experiments. Passaging was performed using 0.05% trypsin-EDTA (Thermo Fisher) for a maximum of 4 min.

Rabbit cardiac fibroblasts were plated onto 24-well uncoated plates (CytoOne, USA Scientific). Once they reached ~30% confluence, rabbit cardiac fibroblasts were treated with rabbit fibroblast media supplemented with either DMSO (Thermo Fisher) vehicle or etoposide (MedChemExpress), replacing with fresh media and drug every 2–3 days. For conditioned media, cells were washed in PBS before addition of 3-week-old cardiomyocyte (3wkRCM) media for 24 hr, after which conditioned media was harvested, centrifuged at 1000 × *g* for 5 min at 4°C, then filtered with a 0.45 µm cell filter(CytoOne). Cells plated and treated in parallel were used to count cells, and aliquots of equal cell density were stored at –80°C. For use, conditioned media was thawed in a 37°C water bath, diluted 1:1 with fresh 3wkRCM media, then immediately treated onto 3wkRCMs.

## Three-week-old rabbit cardiomyocyte isolation

Primary cardiomyocytes were isolated from the LV and septum of 3-week-old female rabbits as previously described (*Cooper et al., 2013*; *Kabakov et al., 2021*). Myocytes were incubated with either SASP-CM, Pro-CM, or UCM (basal media used consisted of DMEM [Life Technologies] supplemented with 7% fetal bovine serum, penicillin [100 µg/mL], and streptomycin [0.01 µg/mL]) for 30 min in a 37°C incubator before patch clamping.

## Electrophysiology recording

### Fibroblasts

Whole-cell patch clamp recordings from proliferating and senescent fibroblasts were performed with an Axopatch-200B amplifier and pCLAMP 10 software (Molecular Devices, San Jose, CA, USA) at 34–36°C. Capacitance and 60–80% of series resistance were routinely compensated. The sampling frequency was 20 kHz, and –3 dB cut-off frequency was 5 kHz. The pipette solution contained 110 mM K-aspartate, 20 mM KCl, 1 mM $MgCl_2$, 0.05 mM EGTA, 10 mM HEPES, 5 mM $K_2ATP$, 0.1 mM Tris-GTP, and 5 mM $Na_2$-phosphocreatine (pH 7.3 with KOH). Tyrode solution was used as a bath solution and contained: 140 mM NaCl, 5.4 mM KCl, 1.8 mM $CaCl_2$, 1 mM $MgCl_2$, 0.33 mM $NaH_2PO_4$, 5.5 mM glucose, and 10 mM HEPES (pH 7.3 with NaOH). Fibroblasts were clamped at –60 mV holding potential followed by a series of 3 s voltage steps from –100 mV to 40 mV. To minimize changes in the intracellular ion concentrations in the fibroblasts, the resting potential was recorded with sharp electrodes (30–40 MΩ) filled with 3 M KCl in the current-clamp mode (I=0) of the amplifier.

## Electrophysiology recording

### Three-week-old rabbit cardiomyocytes

Whole-cell patch clamp recordings from 3wkRCMs were performed with an Axopatch-200B amplifier and pCLAMP 10 software at 34–36°C exactly as described previously (*Kabakov et al., 2021*).

## Co-culture

Glass bottom dishes of 3.5 cm were coated with laminin (Santa Cruz, sc-29012) at room temperature for 45 min. 2.5E5 fibroblasts senesced via etoposide treatment were seeded on 3.5 cm laminin-coated glass bottom dishes a day before the addition of cardiomyocytes. 1.25E5 proliferating fibroblasts were seeded on glass bottom dishes a day before cardiomyocyte seeding to allow to double once in order to reach approximately 2.5E5 cells. DMEM/F-12 (Life Technologies) supplemented with penicillin (100 µg/mL) and streptomycin (100 µg/mL) was used for fibroblasts until the addition of myocytes. Freshly isolated 4-month-old rabbit cardiomyocytes were allowed to gravity pellet for 30 min. The supernatant was removed, and cells were resuspended in serum-free myocyte media (Medium 199, Earle's Salts, Thermo Fisher Scientific). Finally, fibroblast media was removed, and cardiomyocytes were added to dishes containing fibroblasts with 6.2E4 live myocytes per dish. Dishes were kept at 37°C for 24 hr before patch clamping.

## Co-culture electrophysiology

APs were recorded from adult ventricular cardiomyocytes by whole-cell patch clamp in current clamp mode with an Axopatch-200B amplifier (Molecular Devices, San Jose, CA, USA) at 34–36°C. The pipette solution contained (in mM): 100 K-aspartate, 25 KCl, 10 NaCl, 0.01 EGTA, 10 HEPES, 3 Mg-ATP, 0.002 Na-cAMP, 10 Tris-phosphocreatine (pH 7.3 with KOH). The Tyrode bath solution contained (in mM): 140 NaCl, 5.4 KCl, 1.8 $CaCl_2$, 1 $MgCl_2$, 0.33 $NaH_2PO_4$, 5.5 glucose, 10 HEPES (pH 7.3 with NaOH). AP recordings were analyzed with a Python (Python Software Foundation, Beverton, OR, USA) software script written in the lab. The APD was measured from the time of maximum slope during the rising phase of the AP to either 50% or 90% repolarization from the peak amplitude. For APD statistics and plotting, Origin 2019 (OriginLab Corporation, Northampton, MA, USA) was used. IKr and Ito currents were recorded from ventricular cardiomyocytes by whole-cell patch clamp with an Axopatch-200B amplifier at 34–36°C. The pipette solution contained (in mM): 130 KCl, 10 NaCl, 0.36 $CaCl_2$, 5 EGTA, 5 HEPES, 5 glucose, 5 Mg-ATP, 5 Tris-phosphocreatine, 0.25 Tris-GTP (pH 7.3 with KOH). The Tyrode bath solution contained (in mM): 140 NMDG, 5.4 KCl, 1 $CaCl_2$, 1 $MgCl_2$, 0.2 $CdCl_2$, 7.5 glucose, 5 HEPES (pH 7.3 with HCl). The protocol for IKr recordings was a 3 s voltage step protocol (40 to –30 mV, 10 mV intervals) from a holding potential of –40 mV. The IKr protocol was

recorded before and after adding 5 mM E4031. The peak E4031 sensitive tail currents were analyzed. Ito was recorded in the presence of E4031 with a 500 ms voltage step protocol (50 mV to –50 mV, 10 mV intervals) from a holding potential of –70 mV. Capacitance and 60–80% of series resistance were routinely compensated. $I_{Kr}$ and $I_{to}$ recordings were analyzed in Clampfit 10.6 (Molecular Devices, San Jose, CA, USA).

## Rabbit ventricular myocyte AP and fibroblast modeling

We used the rabbit ventricular myocyte AP model of *Mahajan et al., 2008*, coupled to a passive current fibroblast model based on the currents we recorded in patch clamp experiments. The current amplitudes were measured relative to the tail of the current at the end of the voltage step to minimize the inclusion of leak current. The resulting normalized current-voltage relation was plotted for proliferating and senescent fibroblasts and each fit with a Boltzmann equation (see *Figure 9—figure supplement 1*). The Boltzmann curves were shifted to bring the reversal potential to the resting potentials recorded with sharp intracellular electrodes. The resulting equations were used to model the proliferating and senescent fibroblast currents with the addition of gPro and gSen conductance constants multiplied by the equations to allow adjustment of the current amplitude. To calculate the extrapolated capacitance for the proliferating and senescent myofibroblasts used in the model, the surface area was calculated from the mean volumes in *Figure 9A* assuming spherical cells. Then we calculated the ratio of a single fibroblast surface area in tissue to a single fibroblast surface area in culture for proliferating and senescent cells. These ratios were used to scale the capacitance values measured by whole-cell patch clamp (*Figure 9B*) to 'tissue' capacitance values. Since the measured capacitance was similar in proliferating and senescent fibroblasts (probably due to the normalization of cell size with trypsin treatment on the day of recording), we used the tissue-extrapolated capacitance value for proliferating cells and then multiplied that by 1.95, the ratio of senescent to proliferating fibroblast surface area in tissue. The gap junction conductance was set to 20 nS. Numerical calculation of the different equations in the model was performed by the Euler method with an adaptive time step of t=0.05–5 µs. The simulations to investigate the effect of coupling senescent myofibroblasts to ventricular myocytes were carried out with double precision on a Nvidia Tesla K40 multicore graphic processing unit using the CUDA toolkit (https://developer.nvidia.com/cuda-zone). The data visualization was done using Python (Python Software Foundation, Python Language Reference, version 3.8).

## Statistical analysis

Unless otherwise stated, all data are represented as means ± SD, statistics comparing two groups were two-tailed exact tests, and statistics comparing more than two groups were two-way ANOVA with Bonferroni correction. GraphPad Prism 9.1 was used to perform all statistics and create all graphs.

## Acknowledgements

We thank Dr. Alison Chambers for her consultation on biostatistics. We also thank Dr. Christoph Schorl at the Genomics Core of Brown University for his assistance with qPCR hardware and bioanalysis of RNA samples. We thank Dr. Patrycja Dubielecka for her consultation regarding markers of inflammation. We also thank Cindy Phun, Maria Veliz, Tiffany Borjeson, and the rest of the Rhode Island Hospital Central Research Facilities for assistance during rabbit surgeries and coordination and care of rabbits before and after surgery. This work was funded by NIH grants R01HL139467, 1R1AG049608-01 and T35 HL094308. ES is supported by TUBITAK (BIDEB 2214A).

## Additional information

### Funding

| Funder | Grant reference number | Author |
| --- | --- | --- |
| NHLBI Division of Intramural Research | R01HL139467 | John Sedivy Gideon Koren |

| Funder | Grant reference number | Author |
|---|---|---|
| NHLBI Division of Intramural Research | 1R1AG049608-01 | John Sedivy Gideon Koren |
| TUBITAK | BIDEB 2214A | Elif Sengun |
| National Institutes of Health | T35 HL094308 | Eric Mi |

The funders had no role in study design, data collection and interpretation, or the decision to submit the work for publication.

## Author contributions

Brett C Baggett, Conceptualization, Resources, Data curation, Software, Formal analysis, Supervision, Funding acquisition, Validation, Investigation, Visualization, Methodology, Writing – original draft, Project administration, Writing – review and editing; Kevin R Murphy, Conceptualization, Resources, Data curation, Software, Formal analysis, Validation, Investigation, Visualization, Methodology, Writing – original draft, Writing – review and editing; Elif Sengun, Eric Mi, Conceptualization, Resources, Data curation, Software, Formal analysis, Validation, Investigation, Visualization, Methodology, Writing – review and editing; Yueming Cao, Data curation; Nilufer N Turan, Yichun Lu, Lorraine Schofield, Resources, Data curation, Methodology, Writing – review and editing; Tae Yun Kim, Anatoli Y Kabakov, Peter Bronk, Conceptualization, Resources, Data curation, Software, Formal analysis, Investigation, Visualization, Methodology, Writing – review and editing; Zhilin Qu, Conceptualization, Resources, Data curation, Software, Formal analysis, Visualization, Methodology, Writing – review and editing; Patrizia Camelliti, Dmitry Terentyev, Conceptualization, Resources, Supervision, Writing – review and editing; Patrycja Dubielecka, Conceptualization, Resources, Supervision, Methodology, Writing – review and editing; Federica del Monte, Conceptualization, Resources, Data curation, Software, Formal analysis, Supervision, Visualization, Methodology, Writing – review and editing; Bum-Rak Choi, Conceptualization, Resources, Data curation, Software, Formal analysis, Supervision, Funding acquisition, Visualization, Methodology, Project administration, Writing – review and editing; John Sedivy, Conceptualization, Resources, Data curation, Software, Formal analysis, Supervision, Funding acquisition, Investigation, Visualization, Methodology, Writing – original draft, Project administration, Writing – review and editing, Provided critical input in in vitro experiments and overall experimental design; Gideon Koren, Conceptualization, Resources, Software, Supervision, Funding acquisition, Investigation, Methodology, Writing – original draft, Project administration, Writing – review and editing, Supervised all experiments and writing of the article

## Author ORCIDs

Brett C Baggett http://orcid.org/0000-0003-2227-0004
Peter Bronk http://orcid.org/0000-0002-9067-2016
Patrycja Dubielecka http://orcid.org/0000-0003-3987-0647
Gideon Koren http://orcid.org/0000-0002-6211-5837

## Ethics

This investigation conformed with the current Guide for Care and Use of Laboratory Animals published by the National Institutes of Health (NIH Publication, Revised 2011) as well as the standards recently delineated by the American Physiological Society ("Guiding Principles for Research Involving Animals and Human Beings") and was approved by the Institutional Animal Care and Use Committee of Rhode Island Hospital (Permits numbers 5001-21 and 5040-22).

## Decision letter and Author response

Decision letter https://doi.org/10.7554/eLife.84088.sa1
Author response https://doi.org/10.7554/eLife.84088.sa2

# Additional files

## Supplementary files

- Supplementary file 1. Antibodies used for immunofluorescence staining.
- Supplementary file 2. Rabbit-specific primers used for qPCR.

• MDAR checklist

## Data availability

All data generated or analyzed during this study are included in the provided Source Data file.

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
