## [Editor Report]

This study describes important results and convincing evidence linking myofibroblast senescence in the aged heart with a pro-arrhythmogenic phenotype. This is in turn related to higher mortality after myocardial infarction in the aged rabbit heart. These constitute important empiric as opposed to detailed findings. They nevertheless will be of interest to clinician scientists studying cardiac function and disease.

---

## [Decision Letter]

**Decision letter after peer review:**

Thank you for submitting your article "Myofibroblast Senescence Promotes Arrhythmogenic Remodeling in the Aged Infarcted Rabbit Heart" for consideration by *eLife*. Your article has been reviewed by 2 peer reviewers, and the evaluation has been overseen by a Reviewing Editor and Mone Zaidi as the Senior Editor. The following individual involved in the review of your submission has agreed to reveal their identity: Paul Delgado Olguin (Reviewer #1).

Both reviewers have responded positively to the overall science. Reviewer 1 considers that you have presented important results and convincing evidence linking myofibroblast senescence in the aged heart with a pro-arrhythmogenic phenotype and higher mortality after myocardial infarction in the aged rabbit heart. He/she does comment that the evidence supporting the contention that senescent myofibroblasts promote arrhythmia when coupled with cardiomyocytes is incomplete and would benefit from experimentally testing the in-silico model. He also comments that stronger evidence of coupling of myofibroblasts and cardiomyocytes linked to arrhythmia would raise interest from basic and clinician scientists studying cardiac function and disease. Reviewer 2's comments are of a more technical and presentational nature.

Essential revisions:

The reviewers make the following scientific comments which we would request you address:

Reviewer 1's comments concern the detailed interpretation of the results:

1) The cells in Figure 4F and G, do not appear to correspond to either myofibroblasts or endothelial cells. Given that γH2AX+ cells are significantly increased in the aged heart, could the results presented suggest that a different cell type might be more important for the aged heart's response to MI? Providing some insight into the identity of these cells would be helpful to better understand the results presented. For example, cardiomyocyte senescence could contribute to arrhythmic phenotypes.

2) The results presented show that treatment of cardiomyocytes with conditioned media from, and co-cultured with, senescent myofibroblasts did not change action potential duration in cardiomyocytes. This led to the conclusion that paracrine signalling is unlikely to contribute to a pro-arrhythmogenic phenotype. It is possible that cardiomyocytes do couple with myofibroblasts in the in vitro system used. In which case, the results presented would not favor the proposed model.

3) Another important possibility to be considered is that myofibroblasts might not have produced senescence-associated secretory phenotype-mediators at concentrations high enough to alter action potential duration in the conditions tested. Experimental evidence of the levels of selected mediators of the senescence-associated secretory phenotype in conditioned media would help assess a potential paracrine effect.

4) The evidence of coupling, i.e., the presence of connexin-43 in the interphase between αSMA+ and cardiomyocytes needs to be strengthened. Perhaps analyzing Z-stack 3D reconstructions would help to better define adjacent cells and more precisely reveal the localization of connexin-43.

Reviewer 2 has more technical issues to raise:

1) Some reflection on the media used to study paracrine effects is needed – more experiments here would be beneficial.

2) Patch clamp experiments – how does bath solution alter the effect of any limited paracrine effect – we are removing cells from the treatment media and putting them in physiological solutions – an opportunity to recover?

---

## [Author Response]

Essential revisions:The reviewers make the following scientific comments which we would request you address:Reviewer 1's comments concern the detailed interpretation of the results:1) The cells in Figure 4F and G, do not appear to correspond to either myofibroblasts or endothelial cells. Given that γH2AX+ cells are significantly increased in the aged heart, could the results presented suggest that a different cell type might be more important for the aged heart's response to MI? Providing some insight into the identity of these cells would be helpful to better understand the results presented. For example, cardiomyocyte senescence could contribute to arrhythmic phenotypes.

Mouse lineage tracing studies have shown that myofibroblasts persist in the infarct border zone for long periods of time, however they lose aSMA expression up to at least 4 weeks post-MI (Kanisicak 2018 J Clin Invest, PMID 29664017). We suspect the gH2AX+ aSMA- cells are myofibroblasts that have lost aSMA expression. We attempted to verify this using stainings for other myofibroblast-specific markers (including vimentin, PDGFRα, DDR2, and Tcf21), but such stainings were not successful due to limited availability of verified primary antibodies against rabbit tissue for immunofluorescent microscopy. We did not observe evidence of cardiomyocyte senescence through SABgal staining or gH2AX staining.

2) The results presented show that treatment of cardiomyocytes with conditioned media from, and co-cultured with, senescent myofibroblasts did not change action potential duration in cardiomyocytes. This led to the conclusion that paracrine signalling is unlikely to contribute to a pro-arrhythmogenic phenotype. It is possible that cardiomyocytes do couple with myofibroblasts in the in vitro system used. In which case, the results presented would not favor the proposed model.

It is possible that cardiomyocytes do couple electrically with myofibroblasts in the in vitro system used. However, we have found no evidence of coupling in our co-culture assay. We recorded transient capacitive currents during a hyperpolarizing voltage step (-80 to -100 mV) in cardiomyocytes cultured on top of senescent myofibroblasts, as well as in single cardiomyocytes cultured alone. We fit the transient current decay with a two-exponential decay function. If a cardiomyocyte and myofibroblast were coupled, we would expect that the capacitive transient would have a significantly larger slow time constant and greater total capacitive charge (measured as an integral of the capacitive transient) than in a single myocyte in the absence of myofibroblasts. This is because in the coupled cells the total electric capacitance of cell membranes is larger, and the electrical access resistance of gap junctions to the myofibroblast is significantly higher than the pipette access resistance to the cardiomyocyte membrane. However, we found no significant difference in time constants between these two groups. A similar analysis was done to show coupling of descending vasa recta endothelial cells (Zhang 2006 Am J Physiol Regul Integr Comp Physiol, PMID 16840652). Furthermore, when we added 100 mM carbenoxolone to block gap junctions in the co-culture system, there was no significant difference in the slow time constants between the two groups of cells, indicating no significant electrical connections between the cardiomyocytes and the fibroblasts. A sentence mentioning this analysis has been added to the manuscript (lines 468-471).

3) Another important possibility to be considered is that myofibroblasts might not have produced senescence-associated secretory phenotype-mediators at concentrations high enough to alter action potential duration in the conditions tested. Experimental evidence of the levels of selected mediators of the senescence-associated secretory phenotype in conditioned media would help assess a potential paracrine effect.

This is possible, or alternatively the long-term effect of relatively low concentration paracrine signaling, as expected from aged infarcted animals, might have an effect on cardiomyocyte electrophysiology. Previous studies in mice and rats have indicated that exogenous administration of individual cytokines found in the SASP can alter the activity of ion channels and prolong cardiomyocyte APD (Stuart 2016 J Mol Cell Cardiol PMID 26739214). These are important questions to address to fully understand the spatiotemporal contribution of senescence towards inflammation and arrhythmias, however the results presented in this manuscript are not meant to rule out a paracrine effect. Further investigation is needed to characterize nuances in potential paracrine and juxtacrine effects but is outside the scope of this manuscript. Of note, other investigators reported that IL-6 may block I_Kr_ (Chowdhury 2021 Int J Mol Sci, PMID 34681909; Aromolaran 2018 PLoS One, PMID 30521586). Given that the conditioned media had no significant effects on I_Kr_ current density and APD, we feel that assessing the low levels of multiple cytokines in the media will not be productive.

4) The evidence of coupling, i.e., the presence of connexin-43 in the interphase between αSMA+ and cardiomyocytes needs to be strengthened. Perhaps analyzing Z-stack 3D reconstructions would help to better define adjacent cells and more precisely reveal the localization of connexin-43.

We appreciate the reviewer’s suggestion. Surely, the reviewer is aware of several elegant studies by Mahoney et al. (Mahoney 2016 Sci Rep, PMID 27244564), Quinn et al. (Quinn 2016 Proc Natl Acad Sci U S A, PMID 27930302), and Rubart et al. (Rubart 2018 Cardiovasc Res, PMID 29016731) that demonstrated electrical coupling between cardiomyocytes and myofibroblasts (nonmyocytes) in the infarcted mouse heart. The latter report also provided 3D reconstructions of fluorescence confocal images that were acquired from infarct border zones. Here (Figure 6B-D), the authors presented convincing data that showed the expression of connexins 43 and 45 between myocytes and myofibroblasts. As confocal imaging requires a compromise between resolution, scan time, and photobleaching of the samples, we decided to choose a high enough resolution (approximately 200um width, 200um height, 6um depth) to create informative 2D images (the juxtaposition of cardiomyocytes and senescent myofibroblasts). This information was added to the methods section, (lines 878-879). However, the resolution chosen is not high enough for any convincing 3D reconstruction as we felt at that time that an in-depth study of the aforementioned already established heterocellular coupling would not add any new information to the manuscript.

However, we would like to present the reviewer additional confocal images that show the presence of Cx43 between cardiomyocytes and myofibroblasts (indicated by yellow circles) confirming the aforementioned observations made by Rubert et al. in a larger animal. These additional images have been included in the manuscript as Figure 8—figure supplement 1 (line 541). Our data would also somewhat corroborate an earlier finding (Camelliti 2004 Circ Res, PMID 14976125) that reported coupling between neighboring fibroblasts and myocytes in rabbit sinoatrial node as evidenced by dye transfer.

Reviewer 2 has more technical issues to raise:1) Some reflection on the media used to study paracrine effects is needed – more experiments here would be beneficial.

In our experience, rabbit cardiomyocytes deteriorate over course of hours in unusual or substandard media. Standard cardiomyocyte growth media used was DMEM + 7% FBS + 1X Pen/Strep, and standard cardiac fibroblasts growth media used was DMEM-F12 + 10% FBS + 1X Pen/Strep. Standard cardiomyocyte growth media as described in the methods section (lines 952-953) was used to co-culture the myocytes and fibroblasts 24-30 hours before the patch clamping. The discussion was revised to include reflection on how the media used might have masked a paracrine effect (lines 762-763), however a several-hour co-culture in substandard media might induce stress in cardiomyocytes that might also introduce artefacts.

2) Patch clamp experiments – how does bath solution alter the effect of any limited paracrine effect – we are removing cells from the treatment media and putting them in physiological solutions – an opportunity to recover?

Correct, the patch clamping media differed from the co-culture media, although the patch clamping media was introduced shortly before patch clamping. Any possible rapid recovery of cardiomyocytes immediately before patch clamping would therefore not be detected in our experiments. However, we believe that such a scenario is unlikely. This this addressed in the revised discussion (lines 765-767).